# Estimating cognitive biases with attention-aware inverse planning

**Sounak Banerjee**[1,*]   **Daphne Cornelisse**[1]   **Deepak Gopinath**[2]   **Emily Sumner**[2]
**Jonathan DeCastro**[2]   **Guy Rosman**[2]   **Eugene Vinitsky**[1]   **Mark K. Ho**[1]
[1]New York University   [2]Toyota Research Institute
{sounak.banerjee, dc4971, vinitsky.eugene, mark.ho}@nyu.edu
{deepak.gopinath, emily.sumner, jonathan.decastro, guy.rosman}@tri.global

## Abstract

People's goal-directed behaviors are influenced by their cognitive biases, and autonomous systems that interact with people should be aware of this. For example, people's attention to objects in their environment will be biased in a way that systematically affects how they perform everyday tasks such as driving to work. Here, building on recent work in computational cognitive science, we formally articulate the *attention-aware inverse planning problem*, in which the goal is to estimate a person's attentional biases from their actions. We demonstrate how attention-aware inverse planning systematically differs from standard inverse reinforcement learning and how cognitive biases can be inferred from behavior. Finally, we present an approach to attention-aware inverse planning that combines deep reinforcement learning with computational cognitive modeling. We use this approach to infer the attentional strategies of RL agents in real-life driving scenarios selected from the Waymo Open Dataset, demonstrating the scalability of estimating cognitive biases with attention-aware inverse planning.

## 1   Introduction

Conventional wisdom suggests that people are not always rational, and this truism is now backed by over half a century of research in cognitive science [Simon, 1955, Tversky and Kahneman, 1974, Gigerenzer et al., 2000]. It is remarkable, then, that standard approaches to modeling goal-directed human behavior in machine learning and artificial intelligence (e.g., inverse reinforcement learning) assume that people act optimally—or at least approximately optimally—with respect to their goals [Ng and Russell, 2000, Arora and Doshi, 2021]. What explains this disconnect between psychological reality and basic modeling assumptions? How can we bridge this gap?

One reason why machine learning models of human behavior assume (approximate-)optimality is that although this assumption is false in fundamental ways, it underpins *expected utility theory* [von Neumann and Morgenstern, 1947, McFadden, 1974], which itself provides the basis for developing *scalable* algorithms that can make *generalizable* predictions about human behavior [Ho and Griffiths, 2022]. In contrast, general and scalable frameworks that account for why, when, and how people deviate from idealized rationality have not been previously available. This has changed in recent years: Building off of classic ideas in bounded rationality [Simon, 1990], cognitive scientists have begun to go beyond just *describing* deviations from ideal rationality to systematically *explaining*, *predicting*, and *influencing* those deviations by developing general theories of resource-limited cognition [Lewis et al., 2014, Griffiths et al., 2015, Gershman et al., 2015, Shenhav et al., 2017, Bhui et al., 2021]. The present work aims to apply these recent insights to the development of generalizable and scalable algorithms for modeling human decision-making.

39th Conference on Neural Information Processing Systems (NeurIPS 2025).

Here, we focus specifically on how limited attention affects goal-directed behavior in complex tasks. As an example, consider the everyday activity of driving. When driving, people must systematically prioritize which objects in the environment to attend to (e.g., the car directly ahead, the child entering a blind spot) and which objects to ignore (e.g., the parked car on the opposite side of the street, the pile of leaves in a driveway). Whether an object is prioritized for attention depends on numerous factors, such as its immediate relevance for planning or a person's previous driving experience. Recent work by Ho et al. [2022] has proposed the framework of *value-guided construal*, which formalizes the process of prioritizing attention for planning as a meta-cognitive decision. The value-guided construal framework can be viewed as a principled *forward model* of how unobservable cognitive processes (e.g., attention, planning, and biases) integrate to produce observable behavior.

In this paper, we extend the value-guided construal framework to incorporate attentional biases and then invert it to specify the *attention-aware inverse planning* problem, in which the goal is to infer the biases of an attention-limited decision-maker from their behavior. We then study this approach in tabular domains and compare it to standard modeling approaches that assume decision-makers have *un*limited attention (e.g., inverse reinforcement learning [Abbeel and Ng, 2004]). Finally, we develop an approach for inferring biases from synthetically generated behavior in real-world driving scenarios selected from the Waymo Open Motion Dataset [Ettinger et al., 2021], demonstrating the feasibility of scaling up attention-aware inverse planning to complex, naturalistic domains.

## 2 Background

A number of algorithmic approaches have been developed for modeling human behavior. The simplest approaches are based on *imitation learning* [Bain and Sammut, 1995], in which behavior is modeled by directly fitting a policy that predicts actions given observations. Imitation learning is effective when data is abundant and generalization is within-distribution, and has been successfully applied to model human behavior in real-world tasks such as highway driving [Lefèvre et al., 2014, Rudenko et al., 2020]. However, when training data is limited or generalization is out-of-distribution, it is necessary to turn to methods that make stronger assumptions about policy structure. In particular, methods based on *inverse reinforcement learning* (IRL) assume that observed behavior is generated by a policy that is optimal (plus decision-noise) with respect to the true task dynamics and a latent reward function, and attempts to infer the reward function [Abbeel and Ng, 2004, Ziebart et al., 2008].

In human-robot interaction and AI safety, generalizable models of human cognition and action are key for safe and effective interaction [Collins et al., 2024]. Although multi-agent interactions can be modeled without considering human biases via self-play or population-play algorithms [Silver et al., 2017, Jaderberg et al., 2019], agents trained this way perform poorly with people, especially in cooperative tasks [Carroll et al., 2019]. Additionally, accurate representations of how people make decisions are needed for more sophisticated forms of interaction that involve reasoning about the physical capabilities and intentions of human users [Gopinath et al., 2021a], modeling how humans reason about intentions [Dragan et al., 2013], or more complex forms of cooperation [Hadfield-Menell et al., 2016, Kleiman-Weiner et al., 2016]. Separate from this work on behavior modeling, work on attention modeling in the machine learning community has mainly been pursued in the context of specific tasks, such as driving [Kotseruba and Tsotsos, 2022, Gopinath et al., 2021b, Cao et al., 2022]. Although early work used cognitive models that relate attention to behavior [Salvucci, 2006], more recent work is based on data-driven approaches using gaze data and reaction times [Ahlström et al., 2022, Sundareswara et al., 2013, Li et al., 2024, Yang et al., 2020, Xue et al., 2025].

IRL has also inspired computational approaches to studying human social cognition. For example, Baker et al. [2009] proposed that human theory of mind could be conceptualized as *inverse planning*, Bayesian inference over a generative model of rational action. Additionally, as noted in the introduction, a large literature documents human deviations from expected utility theory [Kahneman and Tversky, 1979]. More recent theoretical work (e.g., *resource rational analysis* [Griffiths et al., 2015]) attempts to unify these disparate findings by explicitly modeling people's decision-making as a form of *resource constrained optimization* [Griffiths et al., 2015, Gershman et al., 2015]. Along these lines, Ho et al. [2022] proposed *value-guided construal*, a computational account of how people form simplified but useful world models when planning to make the best use of their limited attentional resources. Follow up work shows how learned biases [Ho et al., 2023] and properties of visuospatial attention [da Silva Castanheira et al., 2025] interact with and bias value-guided construal. The present paper aims to build on these findings and apply them to behavior and attention modeling.

# 3 Attention-aware inverse planning

This section first reviews preliminaries for sequential decision-making and attention-limited decision-making in the value-guided construal framework. We then describe an extension that incorporates attentional biases and invert this model to define the problem of *attention-aware inverse planning*.

## 3.1 Preliminaries

We model sequential decision-making problems as Markov Decision Processes (MDPs), which consist of a tuple $\langle S, A, T, R, \gamma \rangle$, where $S$ is a state space, $A$ is an action space, $T : S \times A \times S \to [0, 1]$ is a transition function, $R : S \times A \to \mathbb{R}$ is a reward function, and $\gamma \in [0, 1)$ is a discount rate [Sutton and Barto, 2018]. Policies are mappings from states to distributions over actions, $\pi : S \times A \to [0, 1]$. The expected discounted future reward of following a policy $\pi$ from a state $s$ defines its *value function*:

$$V(s, \pi) = \sum_a \pi(a \mid s) \left[ R(s, a) + \gamma \sum_{s'} T(s' \mid s, a) V(s', \pi) \right] \tag{1}$$

An *optimal policy* is one that maximizes value at all states: $\pi^* = \arg\max_\pi V(s, \pi)$, for all $s \in S$.

Additionally, we assume an *object-oriented representation* of states [Diuk et al., 2008], in which each state is a subset of objects $\mathcal{O}$ with features $\mathcal{F}$ that can take on values $\mathcal{V}$ that includes scalars and real vectors. That is, define the "sub-state space" corresponding to a subset of objects $O \subseteq \mathcal{O}$ as the set of all maps from features of $o \in O$ to values, $S_O = (O \times \mathcal{F}) \to \mathcal{V}$. Then the full state space is their union: $S = \bigcup_{O \subseteq \mathcal{O}} S_O$. For example, in a driving setting, objects would correspond to vehicles (including the ego vehicle) or obstacles on the road. A state could then be represented as a map, $s = \{(\texttt{car\_1}, \texttt{location}) \mapsto (0.0, 1.0), (\texttt{car\_1}, \texttt{speed}) \mapsto 2.0, (\texttt{car\_2}, \texttt{location}) \mapsto (5.5, 2.4), ...\}$. We assume that if the transition function is defined for a state $s$ with objects $O$, then it is also defined for the *abstracted* state that only includes a subset of objects $O' \subseteq O$, their features, and their values.

## 3.2 Value-guided construals

The current work aims to characterize attention-aware inverse planning, and our first step towards this goal is to specify a generative model of human planning in which attention plays a central role. To specify such a model, we build on existing work in computational cognitive science on resource-limited planning in humans, specifically the *value-guided construal* approach, which provides a domain-general computational framework for how people form simplified mental representations when planning [Ho et al., 2022, 2023].

In the value-guided construal approach, *top-down* attentional processes are modeled as selecting a subset of relevant details (e.g., objects and/or their affordances) that allow a person to plan an effective solution to a task. As a simple but illustrative example, imagine trying to pass another car on the highway when there are two additional cars, one in the passing lane and one parked on the side of the road. Intuitively, the car being passed and the car in the passing lane are relevant to planning, while the parked car is irrelevant. Value-guided construal formalizes this intuition as a cost-benefit calculation: a *construal* of the situation that includes the cars on the road has a higher *behavioral utility* than one that only includes the car being passed because the latter leads to a plan that could lead to worse outcomes (e.g., potentially crashing into the car in the passing lane). Conversely, the construal with the car being passed and the car in the passing lane is also *less representationally complex* than one that includes all three cars, as it represents fewer entities, but it still has the same behavioral utility since the presence of the parked car does not affect optimal behavior. Thus, value-guided construal predicts that a rational, resource-limited decision-maker trading off behavioral utility and representational complexity will form a construal that only includes the two cars on the road—that is, they will only attend to those two cars.

In the current work, we formalize the process of value-guided construal by controlling which objects are included in a state representation. That is, given a true state $s$ composed of objects $O$, a *task construal* is a subset of objects $C \subseteq O$, and the corresponding *construed state* is $s[C] = \{(o, f) \mapsto v \in s \mid o \in C\}$, the subset of the object-feature to value pairs in $s$ that only includes the objects in $C$. Then, we define the *construed policy* as the optimal policy where choices are made with respect

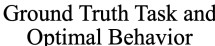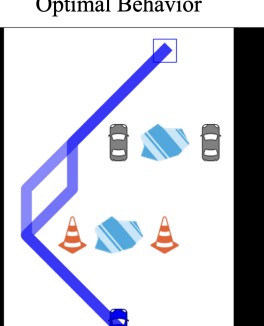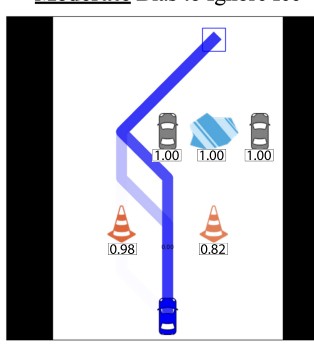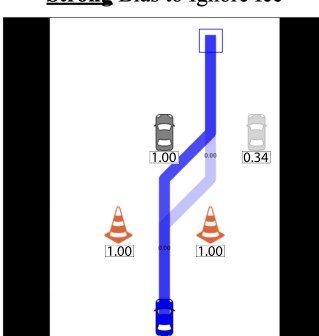

| Ground Truth Task and Optimal Behavior | Construed Task and Behavior with **Moderate** Bias to Ignore Ice | Construed Task and Behavior with **Strong** Bias to Ignore Ice |

Figure 1: Attentional biases affect planned behaviors. (Left) A simple DrivingWorld scenario in which an agent (blue car) receives $+100$ reward upon reaching a goal location (blue square). Hitting a traffic cone results in $-10$ reward, hitting a parked car results in $-100$ reward and termination, and moving on ice leads to slipping to the left or right with probability $0.4$ (see main text for full domain specification). Samples from the optimal policy ($n = 100$, blue lines) avoid both ice patches. (Center) Attention-aware inverse planning assumes that a decision-maker's behavior reflects the formation of simplified task construals that only include task-relevant details [Ho et al., 2022]. In the current scenario, a decision-maker who also has a moderate bias to ignore ice ($\lambda_{\text{Ice}} = -10$) will drive over the first ice patch but not the second. Shown is the average construal and trajectories that the agent expects to occur (that is, it does not consider potentially slipping on the first ice patch). Numbers indicate the marginal probability of attending to an object. (Right) A decision-maker with a strong bias to ignore ice ($\lambda_{\text{Ice}} = -100$) will drive over both ice patches and risk hitting the parked cars. For both, $\lambda_{\text{Cone}} = 10$ and $\lambda_{\text{Parked}} = 0$.

to the construed state: $\pi_C(a \mid s) = \pi^*(a \mid s[C])$.[1] The *value of representation* associated with a construal is then a sum of two terms:

$$\text{VOR}(s, C) = V(s, \pi_C) + \text{Cost}(C) \tag{2}$$

The first term is the *behavioral utility* of a construal, the actual value of acting optimally from a state *as if* the construed state were the case. The second term is the *cognitive cost* of the construal, which we assume is simply the negative of the cardinality of the construal (i.e., the number of objects represented) as in previous work [Ho et al., 2022].

Note that while construals are parameterized as sets of objects, the resulting construed policy $\pi_C$ incorporates all interactions between construed objects. This is because the policy is associated with a value function that is derived from construed reward and transition functions for the task:

$$Q_C(s, a) = R_C(s, a) + \gamma \sum_{s'} T_C(s' \mid s, a) V_C(s') \tag{3}$$

The reward function $R_C(s, a) = R(s[C], a)$ and transition function $T_C(s' \mid s, a) = T(s'[C] \mid s[C], a)$ incorporate the interactions between objects in $C$ (and nothing else). This means the value functions and policy (i.e., $V_C$, $Q_C$, and $\pi_C$) also incorporate information about interactions.

Previous work on value-guided construal focused on simple discrete domains to ensure tight experimental control for human studies. Here, we extend this work in three distinct ways: (1) we incorporate heuristic biases in the construal selection policy, in Section 3.3; (2) we articulate the *attention-aware inverse planning* problem, in which the goal is to infer attentional biases from behavior, in Section 3.4; and (3) we validate the approach in a tabular domain as well as in a driving simulator with continuous states, in Section 4.

---

[1]This formulation of a construed policy in terms of masking states passed to a general optimal policy is roughly equivalent to previous parameterizations (e.g., Ho et al. [2022]), but allows us to use a single pre-trained policy network instead of explicitly computing separate construed policies, as described in Section 4.3.

### 3.3 Modeling biased construal formation

The behavioral utility of a construal formalizes top-down, goal-directed influences on attention, but other factors can influence what construals a person will form. For example, some objects are highly salient due to low-level perceptual features, such as an orange traffic cone or red sports car. Other objects are more or less salient because of heuristics that have been learned from previous experiences in similar tasks. For instance, someone who has extensive experience driving in North Dakota might learn that, by default, one should attend to ice on the road since that has been a frequent type of entity with important consequences for driving. In contrast, someone who primarily drives in southern California may have a weaker bias for attending to ice or even a bias for ignoring ice.

We can extend the original value-guided construal framework to incorporate heuristic strategies by adding a *parameterized bias function*, $H_\lambda(s, C)$ that weights a construal $C$ in a state $s$ and is parameterized by $\lambda$. A person's *construal selection policy* at a state $s$ is then a softmax distribution over the value of representation and the bias [Ho et al., 2023]:

$$\pi_\lambda(C \mid s) \propto e^{\text{VOR}(s,C) + H_\lambda(s,C)} \tag{4}$$

At the beginning of an episode, an attention-limited decision-maker samples a construal according to $\pi_\lambda(C \mid s)$ and then executes its corresponding policy, $\pi_C$, in the environment. Note that here, we do not model dynamic construals or switching construals within an episode, but we return to these factors as important directions for future work in the discussion.

### 3.4 Inferring heuristic biases from behavior

Having specified attention-limited decision making as a form of biased value-guided construal, we can now turn to our main focus: Can we estimate attentional biases from behavior? Of course, in principle, given enough data and an accurate model of a data-generating process, one can estimate the parameters of any stochastic process up to a noise factor. However, it is unclear what this looks like in practice in the context of attention-aware inverse planning. Under what conditions are heuristics going to be identifiable? How much behavioral data is sufficient to obtain accurate estimates? Our goal is to provide preliminary answers to these questions.

Formally, estimating heuristics from behavior can be framed as a probabilistic inference problem. Given a state-action trajectory $\zeta = \langle (s_1, a_1), ..., (s_T, a_T) \rangle$, *maximum-likelihood attention-aware inverse planning* involves finding the set of heuristic bias parameters $\lambda^*$ that maximize the probability of the observed behavior:

$$\lambda^* = \arg\max_\lambda P(\zeta \mid \lambda) \tag{5}$$

$$= \arg\max_\lambda \sum_C P(\zeta \mid C) \pi_\lambda(C \mid s_1) \tag{6}$$

$$= \arg\max_\lambda \sum_C \left[ \prod_t \pi_C(a_t \mid s_t) \right] \pi_\lambda(C \mid s_1) \tag{7}$$

The maximand in Equation 5 can be computed if the space of construals is enumerable and we can compute $\pi_C$. For our experiments in a tabular domain, we calculate these quantities exactly using dynamic programming in JAX [Bradbury et al., 2018] and then solve for $\lambda^*$ using off-the-shelf optimizers [Virtanen et al., 2020]. For experiments involving real-world driving scenarios, we approximate them by using a pre-trained policy network to amortize the computational overhead of planning and masking states to only include subsets of objects.

## 4 Experiments and Results

We conduct three experiments to validate the attention-aware inverse planning approach to behavior modeling. In Section 4.1 we show how it can be used to infer heuristics and attentional biases from behavior in a tabular setting. In Section 4.2, we provide evidence that, unlike our proposed approach, standard approaches like inverse reinforcement learning [Abbeel and Ng, 2004] are not guaranteed to recover attention-limited decision-making. Finally, in Section 4.3 we describe

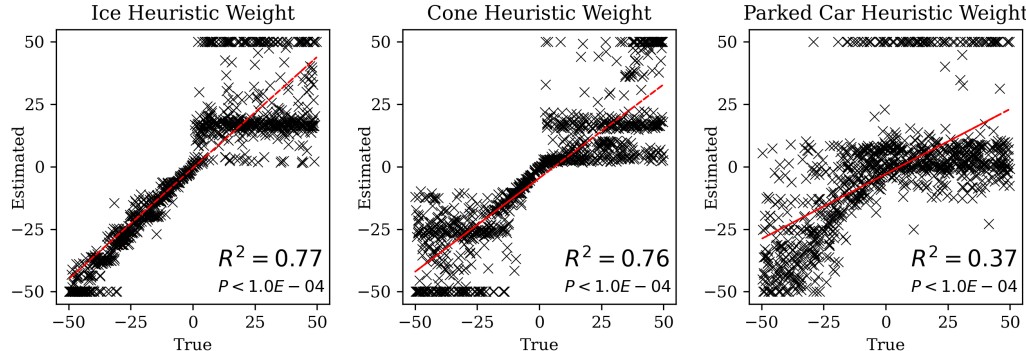

Figure 2: Results of joint maximum-likelihood attention-aware inverse planning: Actual weights of heuristic biases plotted against model estimates, in DrivingWorld (tabular MDP). We simulated the behavior of 1000 agents (with different combinations of weights for three heuristics) by sampling 125 trajectories from 25 different DrivingWorld scenarios (5 from each scenario), for each agent. Then, we attempted to learn the weights jointly from behavior via maximum-likelihood estimation. Each sample represents the true weight (x-axis) of a heuristic for an agent and its estimated value (y-axis) derived from the agent's behavior. Correspondence between true and estimated weights demonstrates that heuristics that are less consequential (e.g., a bias to ignore cones) are more readily identifiable from behavior (indicated by higher $R^2$ values). In contrast, biases that could be more consequential (e.g., a bias to ignore parked cars) are less identifiable because they are outweighed by the effect of value-guided construal. These results illustrate the viability and challenges of attention-aware inverse planning in a tabular setting.

an approach to attention-aware inverse planning in a high-fidelity driving simulator using real-world scenarios. Code for replicating the experiments discussed in this section is available at: `https://github.com/sounakban/gpudrive-CoDec/tree/NeurIPS-2025`.

## 4.1 DrivingWorld

To study how attentional biases affect planned behaviors in a simplified setting, we implemented the tabular DrivingWorld domain shown in Figure 1. The agent controls the blue car and can take the following actions on each timestep: move up 1 cell ($-1$ reward), move up 2 cells ($-1$ reward), move diagonally left ($-2$ reward), and move diagonally right ($-2$ reward). A scenario consists of parked cars, ice patches, traffic cones, and a goal state. The episode terminates if the agent reaches the goal ($+100$ reward), hits a parked car ($-100$ reward), hits a wall ($-100$ reward) or leaves the grid ($0$ reward). Driving into a traffic cone is penalized ($-10$ reward). Finally, there is no additional reward for driving on ice, but with probability $0.4$ the car slips either to the left or right and may inadvertently hit another object.

To apply our model of attention-limited decision-making to a DrivingWorld scenario, we define the bias function $H_\lambda$ using a fixed set of construal feature functions, $\{\phi_i\}$, that map construals and states to real values and parameterize it with linear weights $\{\lambda_i\}$, such that $H_\lambda(s, C) = \sum_i \lambda_i \phi_i(s, C)$. For example, a bias to attend to or ignore ice patches is captured with a weight $\lambda_{\text{Ice}} \in \mathbb{R}$ and feature function $\phi_{\text{Ice}}(s, C)$ that returns the number of ice patches in $C$ that are visible in state $s$. We assume corresponding weights and feature functions for traffic cones ($\lambda_{\text{Cone}}, \phi_{\text{Cone}}$) and parked cars ($\lambda_{\text{Parked}}, \phi_{\text{Parked}}$). To illustrate the effect of different biases, Figure 1 compares the optimal, attention-*un*limited behavior in the task to the behavior of attention-limited decision-makers with a moderate bias to ignore ice ($\lambda_{\text{Ice}} = -10$) and a strong bias to to ignore ice ($\lambda_{\text{Ice}} = -100$).

**Inferring heuristic biases.** To validate our approach for attention-aware inverse planning in DrivingWorld, we apply the procedure to a synthetic dataset generated by first sampling 1000 values of $\lambda_{\text{Ice}}$, $\lambda_{\text{Cone}}$, and $\lambda_{\text{Parked}}$ uniformly sampled between $-50$ and $50$, and then sampling 125 construals and trajectories across the 25 scenarios (see Appendix A). We then computed the joint maximum likelihood heuristic weights, $(\lambda^*_{\text{Ice}}, \lambda^*_{\text{Cone}}, \lambda^*_{\text{Parked}})$. A comparison of the true and estimated values (Figure 2) shows that some parameters can be robustly recovered across a range of values (e.g., $\lambda_{\text{Ice}}$ and $\lambda_{\text{Cone}}$) whereas others are more difficult to recover accurately (e.g., $\lambda_{\text{Parked}}$).

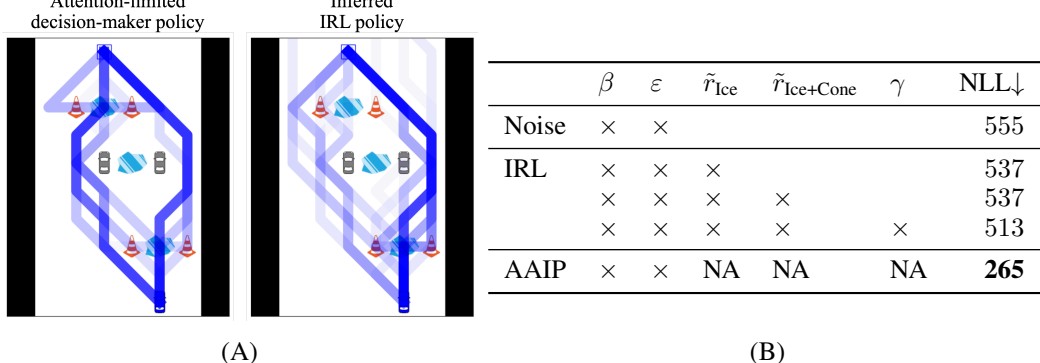

|  | $\beta$ | $\varepsilon$ | $\tilde{r}_{\text{Ice}}$ | $\tilde{r}_{\text{Ice+Cone}}$ | $\gamma$ | NLL$\downarrow$ |
|---|---|---|---|---|---|---|
| Noise | $\times$ | $\times$ | | | | 555 |
| IRL | $\times$ | $\times$ | $\times$ | | | 537 |
| | $\times$ | $\times$ | $\times$ | $\times$ | | 537 |
| | $\times$ | $\times$ | $\times$ | $\times$ | $\times$ | 513 |
| AAIP | $\times$ | $\times$ | NA | NA | NA | **265** |

(A)                                                      (B)

Figure 3: Inverse reinforcement learning (IRL) is not guaranteed to recover attention-limited decision-making. (A) We simulated a ground-truth attention-limited decision-maker with a bias to ignore ice ($\lambda_{\text{Ice}} = -10$) and generated 100 trajectories in the scenario above. This data was passed to an IRL procedure in which we fit the action inverse temperature ($\beta$), random action selection ($\varepsilon$), auxiliary reward for driving on ice ($\tilde{r}_{\text{Ice}}$), auxiliary reward for driving on ice between cones ($\tilde{r}_{\text{Ice+Cone}}$), and discount rate ($\gamma$). We simulated 100 new trajectories in the same scenario with a policy using the fit parameters. Note that compared to the original policy, the resulting IRL policy is noisier and visits different ice patches. (B) As a quantitative evaluation, we compared fitted negative log-likelihoods (NLL) for a policy given optimal action values with fitted decision-noise parameters $\beta$ and $\varepsilon$ (Noise), three increasingly expressive versions of our IRL procedure, and attention-aware inverse planning (AAIP). Noise and AAIP are intended to serve as a ceiling and floor on negative log-likelihoods, respectively. Even when $\beta, \varepsilon, \tilde{r}_{\text{Ice}}, \tilde{r}_{\text{Ice+Cone}}$, and $\gamma$ are fit, IRL has limited ability to capture the data. In the table, $\times$ indicates that a parameter was fit, otherwise it was set to its true value.

## 4.2   Comparison with inverse reinforcement learning

Attention-aware inverse planning generalizes models of inverse decision-making, but a natural question to ask is whether existing approaches can capture regularities in behavioral data even if they mis-specify the data-generating process (e.g., assume people are not attention-limited when in fact, they are). Inverse reinforcement learning (IRL) typically assumes that an expert acts based on an accurate and complete representation of the task dynamics but optimizes an unknown reward function [Abbeel and Ng, 2004]. IRL aims to infer this unknown reward function. Crucially, even though IRL applied to human-like behavior (incorrectly) assumes a decision-maker has unlimited attention, a sensible reward function may always exist that sufficiently captures human behavior.

To demonstrate that this is not always the case, we identified a simple DrivingWorld scenario (Figure 3A) in which inferring rewards is not sufficient to capture the behavior of an attention-limited policy. We first generated 100 trajectories from an attention-limited policy with $\lambda_{\text{Ice}} = -10$, $\lambda_{\text{Cone}} = 10$, and $\lambda_{\text{Parked}} = 0$. To simplify the IRL problem, we assumed that the true reward function is known (hitting cones and parked cars is penalized, reaching the goal is incentivized) but that there is an unknown auxiliary reward added to the true reward. This auxiliary reward added a bonus $\tilde{r}_{\text{Ice}} \in \mathbb{R}$ for visiting any ice patch and $\tilde{r}_{\text{Ice+Cone}} \in \mathbb{R}$ for visiting an ice patch in between cones. We then calculated a combination of the auxiliary rewards parameters, softmax inverse temperature, $\varepsilon$ random choice, and discount rate that maximizes the log-likelihood of the generated data.

As shown in Figure 3A, the IRL policy that best captures the data misses out on several key properties of the original policy. First, it is much noisier and sometimes even fails to reach the goal altogether. Second, whereas the original policy visits the two ice patches between the cones in roughly equal proportion, the IRL policy only visits the first patch and largely avoids the second. We further confirmed that the IRL policy substantially deviates from the original policy with a quantitative comparison of the negative log-likelihoods (Figure 3B).

## 4.3   GPUDrive: A high fidelity driving simulator with real-world data

Next, we turn to methods for scaling up to more complex, continuous domains. Specifically, we test an approach to computing the quantities required for attention-aware inverse planning that involves

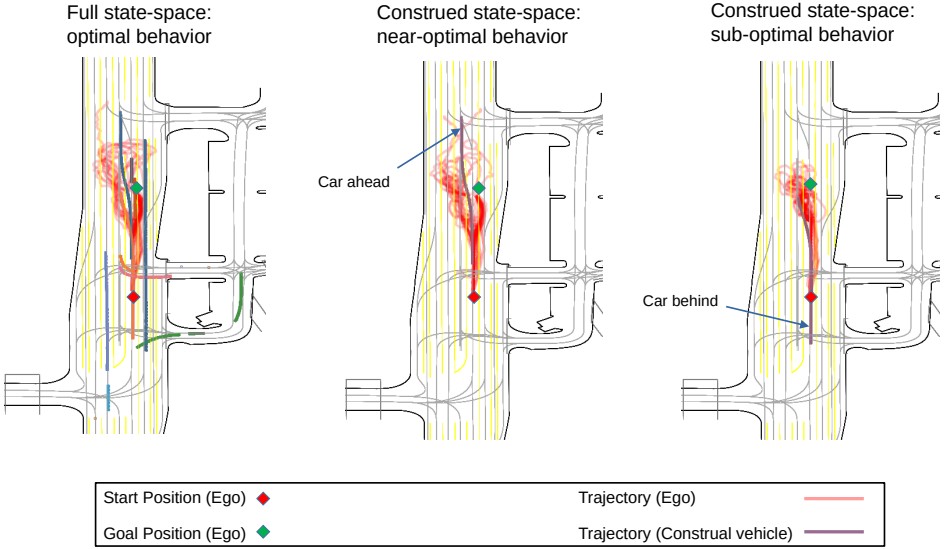

Figure 4: This demonstrates the driving behavior of a PPO agent controlling the ego vehicle in a Waymo highway scenario under three conditions: (i) when it can see every vehicle in the scene (left; optimal); (ii) when it can see a single vehicle ahead, that it needs to pass before it reaches the goal (middle; near-optimal); and (iii) when it can see a single vehicle at its rear that has no impact on the agent's plan (right; sub-optimal). The trajectories, in translucent red, represent the behavior of the ego vehicle controlled by the PPO agent across 40 trials in each condition. Solid lines of all other colors represent the trajectories of other vehicles in the scene. The purple solid lines in the middle and right plots show the trajectories of the single vehicles being observed in each of the two conditions. The behavior of our PPO agent (distribution of ego trajectories) in the optimal (left) and near-optimal (middle) conditions are similar, where the agent goes around the vehicle ahead. In the sub-optimal condition (right), the agent attempts to drive straight towards the goal position, resulting in a crash.

passing masked states, $s[C]$, to a pre-trained *generalist policy* representing $\pi^*$ in order to obtain and evaluate construed policies $\pi_C$. This approach effectively amortizes the cost of computing construed policies, enabling us to scale up to more complex and naturalistic tasks, such as driving.

For our experiments, we use the GPUDrive driving simulator [Kazemkhani et al., 2025] to generate synthetic trajectory data in scenes obtained from the Waymo Open Motion dataset [Ettinger et al., 2021]. Vehicle trajectories were first generated using a biased generative model parameterized by heuristic bias weights $\{\lambda_H\}$. We then showed that it is possible to retrieve the true $\lambda_H$, by performing inference over the synthetic trajectory data, as described in equation 5. We describe the synthetic data generation and inference processes in greater detail in Sections 4.3.1 and 4.3.2, respectively.

### 4.3.1 Synthetic data generation

Ten scenarios were manually selected (illustrations in Appendix B) from the Waymo Open Motion dataset, where the movement trajectory of the agent-controlled (ego) vehicle presented high degrees of interaction with other vehicles in the scene. As a result of our selection process, a large portion of our simulation dataset contains scenarios where the ego vehicle yields to other vehicles, merges into highways, or changes lanes on multi-lane roads. These are all cases where failing to attend to the relevant objects can lead to catastrophic outcomes (crashes).

**Behavioral Utility.** For each scenario, we computed the behavioral utility, $\hat{V}(s, \pi_C)$, of all possible single-vehicle construals (any one vehicle is observed in each construal) using Monte-Carlo estimation. The behavioral utility of a construal was computed by taking the average reward earned by a construed policy—the generalist policy that received a masked (i.e., construed) state that only included a subset of the vehicles from the true state—across $N$ rollouts in the true state (algorithm pseudocode available in Appendix D). For example, if there were 15 vehicles in a scene, then 15 different construed models

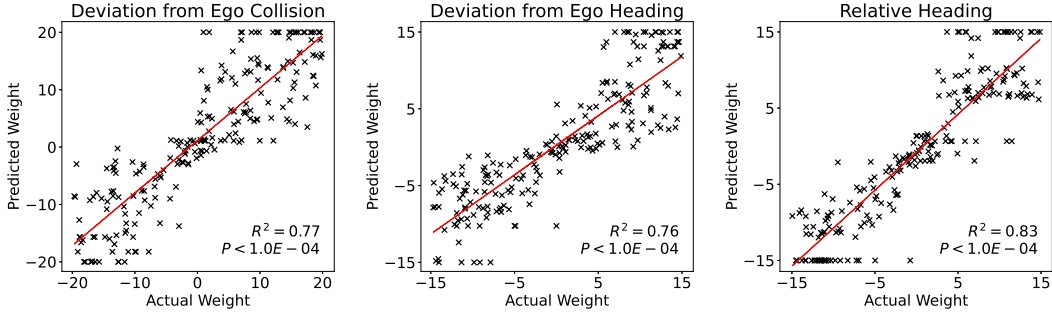

Figure 5: Attention-aware inverse planning results with Waymo Open dataset scenarios in GPU-Drive [Ettinger et al., 2021, Kazemkhani et al., 2025]. We simulated 215 agents with different heuristic weights by sampling 80 trajectories from 10 scenarios per agent. Maximum-likelihood weights were calculated using Bayesian Optimization [Nogueira, 2014]. Each points represents the true weight of a heuristic for an agent and its estimated value derived from the agent's behavior. Correspondence between true/estimated weights demonstrates that this approach can recover underlying agent biases in complex domains such as real-world driving scenarios.

were each rolled out $N$ times, and the behavioral utility of each construal was computed as the average reward of the construed model across these rollouts:

$$\hat{V}(s, \pi) = \frac{1}{N} \sum_{i=1}^{N} \sum_{t=0}^{T(i)} R(s_t^{(i)}, a_t^{(i)}) \tag{8}$$

The value of representation was then obtained for each construal based on Equation 2.

**Heuristic Biases** We computed a construal selection policy, $\pi_\lambda(C \mid s)$, that incorporated three heuristics encoding information about the relative motion of objects, which are features that have been shown to be associated with hazard perception in driving [Lacherez et al., 2014]. The biases were: (i) Deviation from Ego Heading (DfEH), (ii) Relative Heading (RH), and (iii) Deviation from Ego Collision (DfEC) (also see Appendix C). The heuristics were computed at the beginning of each episode, similar to the procedure described in Section 3.3. To perform inference over the heuristic weights of agents for the 10 scenes, we needed ground-truth data containing state information and corresponding agent actions across entire trajectories. However, the Waymo dataset only provides state representations. So, we used a deep-RL PPO model [Cornelisse et al., 2025] pre-trained on GPUDrive as a generalist model to generate sample state-action trajectories $\{\zeta\}$ in the simulator. All other vehicles in the scene followed the logged (human) trajectories from the dataset.

**Generative Model** To generate a trajectory dataset that represents the behavior of an agent with a specific set of parameter values for the three biases $\{\lambda_{\text{DfEH}}, \lambda_{\text{RH}}, \lambda_{\text{DfEC}}\}$, a biased construal selection policy for the specific set of $\lambda$-values was first used to sample eight construals (with replacement) from each of the 10 scenes. Finally, state-action trajectories were generated by rolling out our generalist policy on construed (masked) state-spaces, based on the 8 sampled construals for each of the 10 scenes. This 80-trajectory dataset then served as the synthetic 'ground-truth' behavioral data. 80 trajectories equate to 12 minutes of driving data, which was found to be the minimum amount of data necessary for our algorithm to converge (more details in Appendix G). All simulations were run on an Ubuntu High Performance Computing cluster. A single RTX8000 GPU and 40GB of memory were allocated for the process.

### 4.3.2 Results: inferring heuristic parameters

To validate our approach, we performed a procedure similar to that described in Section 4.1: We uniformly sampled 215 sets of $\lambda$-values for the three heuristics, within a reasonable range of values for each heuristic. For each set, we then sampled 80 trajectories using the generative model described above. Finally, we performed maximum likelihood estimation using Bayesian optimization [Nogueira,

2014] to recover the true $\lambda$s (algorithm available in Appendix E). The results of our inference process are presented in Figure 5, which demonstrate that our algorithm for attention-aware inverse planning can reliably recover underlying heuristic biases of simulated agents in real-world scenarios. Details about program execution latency for the current experiment can be found in Appendix F.

## 5    Limitations and future work

We study attention-aware inverse planning and show how it can scale to continuous, naturalistic domains such as driving, but there are limitations to the current work. First, we constrain the decision-maker to a single fixed construal, chosen at the beginning of an episode, rather than modeling dynamic changes in construal. This can be addressed in future studies by incorporating dynamic updating of task construals in response to task-related events, such as the sudden appearance of new objects. Second, we focus exclusively on how construal leads to systematic deviations from idealized rationality, but people's decisions could be shaped by a variety of other cognitive biases [Lieder and Griffiths, 2020, Schulze et al., 2025] or perceptual mechanisms [Yang et al., 2020]. Future work will need to incorporate these factors. Third, the current work assumes a pre-specified set of possible attentional heuristics, which can be useful for interpretability and data efficiency. However, in situations where interpretability is not a priority or domain knowledge is limited, but behavioral data is abundant, it will be important to extend our approach to learning parameterized heuristics.

Additionally, our work could be extended to incorporate joint inference about rewards and attentional biases since here we assume a known reward model. Previous work in cognitive science has explored how people make these joint inferences [Rane et al., 2023], and theoretical work on "robust IRL" has shown that mis-specification of an expert's dynamics model can lead to systematic error between the value between policies estimated by IRL and the expert policy [Xu and Liu, 2024]. Indeed, the results of section 4.2 suggest that it is not difficult to find scenarios where this error is meaningful and that the recovery gap may be significant in practice. Finally, we apply attention-aware inverse planning in real-world scenarios taken from the Waymo Open Dataset but estimate heuristics from synthetic data. While this approach provides insight into the feasibility of estimating heuristics in complex tasks, validation of our algorithm on human behavior in more tasks is an important next step.

## 6    Conclusion

This paper aims to bridge recent insights on resource-limited decision-making from cognitive science [Griffiths et al., 2015, Gershman et al., 2015, Lieder and Griffiths, 2020] with scalable human behavior modeling [Lefèvre et al., 2014, Rudenko et al., 2020, Abbeel and Ng, 2004, Ziebart et al., 2008]. Specifically, we extend the framework of *value-guided construal* [Ho et al., 2022, 2023], a formal account of how people allocate limited attention during planning, and articulate the *attention-aware inverse planning* problem, in which the goal is to estimate, from a person's actions, what attentional biases influence their behavior. We conducted two sets of experiments to validate this approach to behavior modeling. Our first set of experiments demonstrated the feasibility of estimating cognitive biases from data in a tabular setting and show that standard inverse reinforcement learning can sometimes fail to capture the behavior of attention-limited decision-makers. Our second set of experiments further validate this approach in a high-fidelity driving simulator [Kazemkhani et al., 2025] using real-world scenarios taken from the Waymo Open Dataset [Ettinger et al., 2021].

Aside from demonstrating the feasibility of scaling up attention-aware inverse planning to complex domains such as driving, the current work shows how principled modeling approaches from computational cognitive science [Griffiths et al., 2024] can integrate effectively with modern machine learning methods (e.g., deep RL). In doing so, it contributes both to the development of safe, interpretable, and trustworthy systems that interact with people [Ho and Griffiths, 2022] as well as to the creation of generalizable cognitive models and theories that can capture the full range of natural human behavior [Wise et al., 2024, Carvalho and Lampinen, 2025].

## Acknowledgements

This work was supported by NSF Award #2348442 and the TRI University Research Program.

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

# A  Estimating heuristics in tabular setting

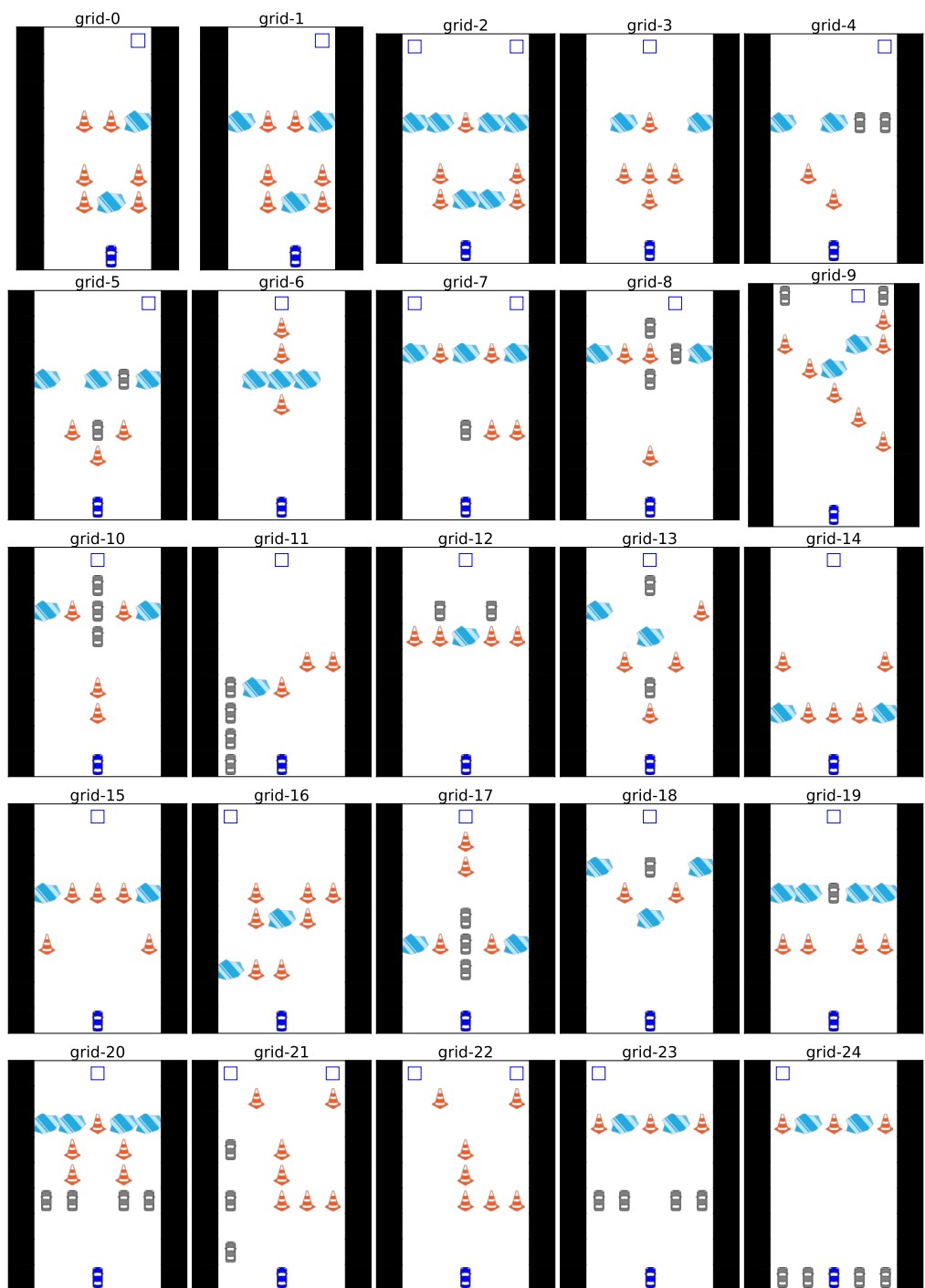

Figure 6: DrivingWorld scenarios used for estimating heuristics in a tabular setting. The true reward function and dynamics are the same as those described in Figure 1.

# B  Waymo Open Dataset scenes selected for our experiment

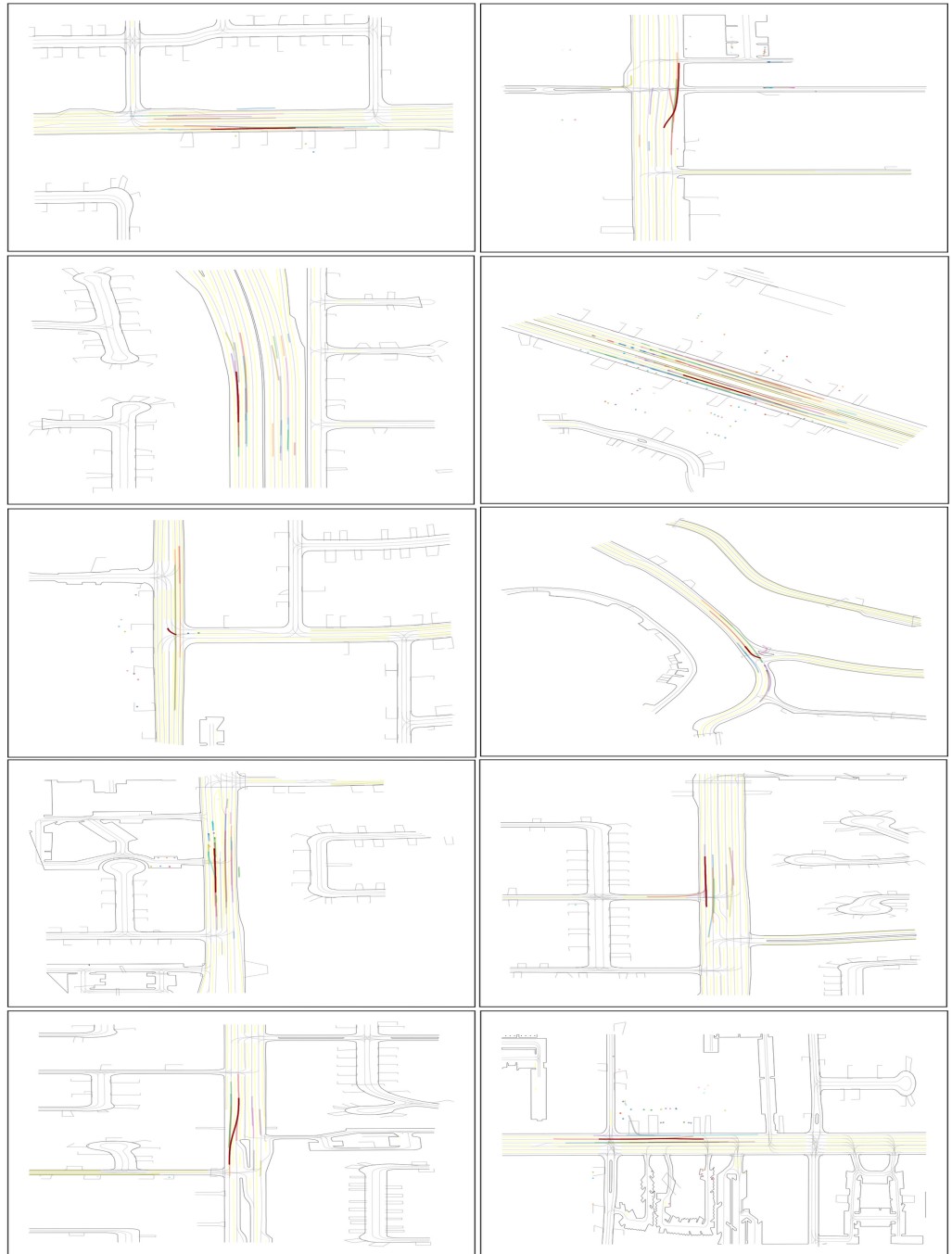

Figure 7: Trajectory illustrations for the 10 scenes chosen from the Waymo dataset. Trajectories marked in dark red represent the movement of the ego vehicle as logged in the original dataset. Other colors represent the movements of other vehicles in the scene.

# C GPUDrive Heuristic Biases

## C.1 Deviation from Ego Heading bias

The DfEH bias $\{\phi_{DfEH}\}$ was computed by taking the average angle (in radians) between the direction of heading of the ego-vehicle and the positional direction of other vehicles (relative to the ego) in the construal.

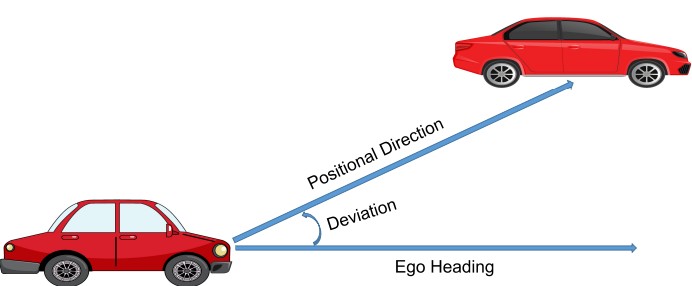

Figure 8: Visual representation for the calculation of Deviation from Ego Heading bias $\{\phi_{DfEH}\}$. NOTE: Car cliparts obtained from freepik

## C.2 Relative Heading bias

The RH bias $\{\phi_{RH}\}$ was computed by taking the average angle (in radians) between the direction of heading of the ego-vehicle and all other vehicles in the construal.

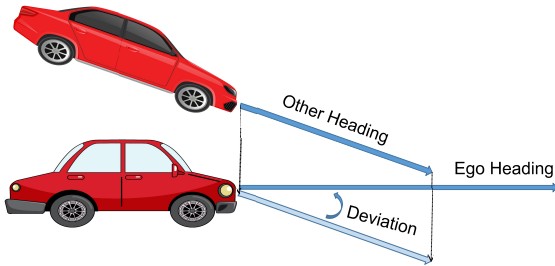

Figure 9: Visual representation for the calculation of Relative Heading bias $\{\phi_{RH}\}$. NOTE: Car cliparts obtained from freepik

## C.3 Deviation from Ego Collision bias

The DfEC bias $\{\phi_{DfEC}\}$ was computed by taking the average value of the dot product of the relative velocity and the relative displacement of each vehicle in the construal, relative to the ego:

$$\phi_{DfEC} = \frac{1}{N} \sum_{N}^{i=1} (V_i - V_e) \cdot (Pos_i - Pos_e) \qquad (9)$$

Where, $N$ is the number of vehicles in a construal; $V_i$ and $V_e$ represent the velocities of the current vehicle and the ego vehicle, respectively; and $Pos_i$ and $Pos_e$ represent the positions of the current and ego vehicles, respectively. Values for this bias were normalized to [-1, 1]. Where 1 indicates that vehicles in the construal are heading towards the ego vehicle, while a value of -1 indicates that vehicles are moving further away from the ego.

## D Algorithm: behavioral utility

The algorithm block summarizes our approach for computing the behavioral utility (VOR) of each construal in each scene. The construed model policy $\pi_C$ is obtained by masking the state-space of the optimal baseline model.

---

**Algorithm 1** Behavioral Utility Calculation

---

**for** scene in all_scenes **do**
    $s \leftarrow S_0(scene)$                 ▷ Start state for the scene
    **for** $C$ in construals($s$) **do**          ▷ Loop through list of construals given $s$
        $N \leftarrow 40$
        $construal\_rewards \leftarrow empty\_list()$
        $\pi_C \leftarrow \pi(C)$               ▷ Get construed model policy
        **while** $N \neq 0$ **do**
            $reward \leftarrow simulate(s, \pi_C)$
            $construal\_rewards.insert(reward)$
            $N \leftarrow N - 1$
        **end while**
        $behavioral\_utilities(scene, C) \leftarrow Average(construal\_rewards)$
    **end for**
**end for**

---

# E Algorithm: $\lambda_H$ inference

The algorithm block describes our approach for inference of heuristic biases ($\lambda_H$). The estimation is performed by maximizing the probability of observing a set of sampled trajectories over values of $\lambda_H$. The maximization in our case was performed using a Bayesian optimization technique for GPUDrive, and a gradient-based optimizer for the DrivingWorld paradigm.

---

**Algorithm 2** Inference Logic

---

**procedure** $Get\_P_{traj|\lambda}$(sampled_trajs, $\lambda_{arg}$)
    $\lambda_{arg}\_list \leftarrow empty\_list()$
    **for** $\zeta$ in sampled_trajs **do**
        $\zeta\_list \leftarrow empty\_list()$
        $s \leftarrow S_0(\zeta)$                                    $\triangleright$ Start state for trajectory $\zeta$
        **for** $C$ in construals($s$) **do**
            $\pi_C \leftarrow \pi(C)$                           $\triangleright$ Get construed model policy
            $P_{\zeta|C} \leftarrow \prod_{\langle s_t,a_t\rangle \in \zeta} \pi_C(a_t \mid s_t)$
            $P_{C|\lambda_{arg}} \leftarrow e^{VOR(s,C)+H_{\lambda_{arg}}(s,C)}$
            $\zeta\_list.insert(P_{\zeta|C} * P_{C|\lambda_{arg}})$
        **end for**
        $P_{\zeta|\lambda_{arg}} \leftarrow \sum \zeta\_list$
        $\lambda_{arg}\_list.insert(P_{\zeta|\lambda_{arg}})$
    **end for**
    $P_{sample|\lambda_{arg}} \leftarrow \prod \lambda_{arg}\_list$
    return $P_{sample|\lambda_{arg}}$
**end procedure**

    sampled_trajs $\leftarrow simulate(\pi_{\lambda_{True}})$        $\triangleright$ Sample trajectories for agent with $\lambda_{True}$ heuristics
    $\lambda_{init} \leftarrow sample(\{\lambda_{DfEH}, \lambda_{RH}, \lambda_{DfEC}\})$            $\triangleright$ Randomly initialize $\lambda$
    $\lambda_{est} \leftarrow maximize_\lambda \ Get\_P_{traj|\lambda}$(sampled_trajs, $\lambda_{init}$)            $\triangleright$ Estimate $\lambda$

---

# F Time Complexity

In this section, we report our algorithm execution time for our experiments with GPUDrive. This is because the data generated by the simulator is a good approximation of real-world high-dimensional complex data.

We had to overcome several computational bottlenecks to perform inference on real-world Waymo data using GPUDrive. The complete pipeline involves: (1) computing an optimal policy, (2) computing behavioral utilities of construals by simulating the optimal policy is construed observation spaces, and (3) performing maximum likelihood estimation over continuous multi-dimensional lambda values.

One of the algorithmic contributions of our paper is a more efficient way to compute the optimal policy, which essentially involves amortizing the cost by pre-training a generalist policy using standard RL methods. This can be done on a standard GPU within a day. GPUDrive includes a pre-trained generalist policy that we used for our experiments.

Computation of behavioral utilities requires simulating a masked policy on the true dynamics. It took an hour to evaluate 166 construals across 10 Waymo scenarios by sampling 40 trajectories per construal, on a Ubuntu High Performance Computing cluster with a single RTX8000 GPU and 40GB of RAM allocated to the process. We were only able to partially parallelize the evaluation process due to limitations in the current implementation of the Python interface, but we note that evaluation can be significantly sped up by optimally parallelizing the process.

Finally, we performed inference over heuristic parameters on a laptop computer (with a Intel Core Ultra 7 165H CPU and 32GB of RAM). It took about 3 min and 30 sec to perform inference over eighty 9-second trajectories (roughly 12 minutes of driving data) for a single set of parameters. Our

current implementation of the inference algorithm is a sequential CPU-based process, which can be parallelized and executed on GPUs.

# G    Sample Efficiency

To test the sample efficiency of our approach in GPUDrive, we evaluated the performance of our algorithm on a range of dataset sizes (expressed as durations of observed real-world driving data), from ten 9-second trajectories (the equivalent of 1.5 minutes) to five hundred 9-second trajectories (75 minutes). Evaluation of performance for different durations was measured over 30 randomly sampled lambda values. Our findings from this experiment are reported in Figure 10.

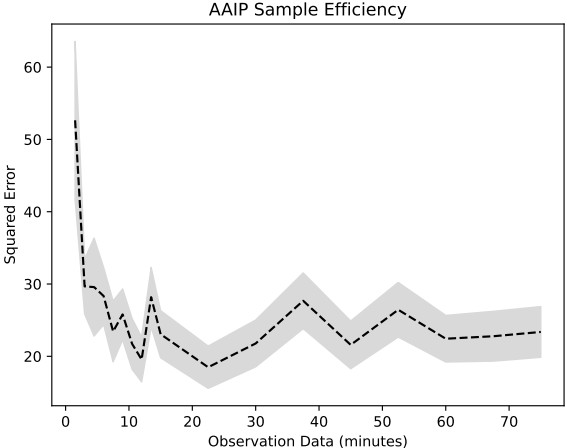

Figure 10: Plot for the efficiency of the AAIP algorithm in estimating true values of lambda with different sample sizes of observation data. The x-axis represents the amount of observation data (in minutes) used for the inference process, while the y-axis represents the mean (and standard error) squared error between true and estimated lambdas. The trend suggests that the algorithm converges at around 12 minutes of observed driving (eighty trajectories).

