# OpenReview forum: "Estimating cognitive biases with attention-aware inverse planning"
_NeurIPS.cc/2025/Conference — NeurIPS 2025 spotlight_

### Official Review · Reviewer_kdHu · 2025-06-27

**Clarity:** 3
**Significance:** 2
**Originality:** 3
**Rating:** 4
**Confidence:** 4

**Summary:**

This article investigates the interesting topic of how goal-directed actions are influenced by human cognitive biases. Specifically, it examines how limitations in human attention lead to biases in attending to objects, which in turn can significantly affect the actions performed.

**Questions:**

Please kindly visit the Strengths And Weaknesses. I do not have further questions.

**Ethical Concerns:**

["NO or VERY MINOR ethics concerns only"]

**Final Justification:**

Based on the authors’ rebuttal, my main concerns have been addressed, and I have accordingly increased my initial score.

**Limitations:**

Yes.

**Paper Formatting Concerns:**

To the best of my knowledge there are not major formatting issues in this paper.

**Quality:**

3

**Strengths And Weaknesses:**

**Strengths**
- The presentation of the main ideas is clear and consistent.
- The main research ideas are elaborated well in the introduction and background, and further in Section 3.
- Experiments are conducted on a synthetic dataset generated using the GPUDrive driving simulator, serving as a proof of concept.


**Weaknesses**
- The paper conducts only limited experiments, all performed on a synthetic dataset. While this serves as a reasonable proof of concept, it leaves unanswered how the proposed ideas would perform in real-world scenarios.

- From what I understand, the experiments use only a single dataset, the Waymo Open Motion dataset, which further limits the work’s generalizability.

- There are no comparisons with related works presented in the paper. The authors state that *“we build on existing work in computational cognitive science,”* but it remains unclear how the proposed model differs from prior work, both in theoretical foundations and applied aspects — for example, in relation to resource-limited planning in humans, such as the value-guided construal model.

- The paper would benefit from a dedicated section for experiments and results. Currently, the presentation of experiments is scattered, making it difficult to follow. I strongly recommend adding a clear Methodology section to describe all aspects of the proposed model, followed by distinct sections for Experiments and Results.

- Additionally, the manuscript lacks a Conclusion section. I suggest including one, even if brief, to summarize key findings and outline future work.

- Figure 2 is difficult to interpret. There does not appear that the caption explaining the figure or how to extract information from it. Additionally, it is unclear how important it is for the reader to understand the estimated heuristic weights in tabular MDPs. Finally, the meaning of the $R^2$ values shown within the figure is not explained.

- Figure 4 is not sufficiently descriptive. The arrows and annotations do not further clarify the context or content depicted, making it difficult to interpret the figure.

---

> ### Author Rebuttal · Authors · 2025-07-31
>
> We thank the reviewer for their interest in our ideas and their thoughtful critiques. We have attempted to address the weaknesses they note in the working manuscript and believe that this has substantially improved the presentation of our work as well as its limitations.
>
> > The paper conducts only limited experiments, all performed on a synthetic dataset. While this serves as a reasonable proof of concept, it leaves unanswered how the proposed ideas would perform in real-world scenarios.
>
> We agree with the reviewer that there is still much work to do applying the proposed ideas to real-world scenarios and now note this more explicitly in the paper. To be clear, however, our experiments are not purely synthetic since the Waymo dataset we used are all real-world scenarios. The ego vehicles in those scenarios are controlled by a synthetic PPO model, but they have to contend with other vehicles that follow trajectories logged from the driving behavior of humans. All aspects of the scenarios, apart from the trajectory of the ego vehicle, represent real-world complexity and behavior. We believe this demonstrates that our approach is compatible with the physical constraints and affordances involved in the real-world task of driving.
>
> That said, related to the concerns of the reviewer regarding real-world applicability, we have also performed a followup study to approximate the real-world data efficiency of our algorithm across data amounts ranging from the equivalent of 1.5 minute to 75 minutes. With 1.5 minutes of driving data, the mean squared error between true and estimated parameters is 52.64 (averaged over 30 simulations). At 12 minutes of driving data this decreases to 19.56 and stays around this value with more data. A plot of these results is now included in the supplement of the manuscript (we did not include it here due to the rebuttal rules). We believe this provides a meaningful demonstration of data efficiency, and it does so precisely because we committed to testing our algorithm in simulations of real-world driving scenarios.
>
> > From what I understand, the experiments use only a single dataset, the Waymo Open Motion dataset, which further limits the work’s generalizability.
>
> We agree with the reviewer that testing on more datasets and domains would be very valuable. That said, we would like to point out that the Waymo open-motion dataset includes 574 hours of driving-data collected from a wide range of driving contexts and scenarios, and it is a standard dataset, widely used for validation of Machine Learning approaches for driving. For our experiment, we further selected scenarios that involve executing some of the most complex driving maneuvers, which include merging, yielding, and passing other cars. We believe that our algorithm’s ability to reliably estimate heuristics in such complex driving contexts in fact demonstrate the versatility of our approach, at least within the domain of  real-world driving situations.
>
> > The authors state that “we build on existing work in computational cognitive science,” but it remains unclear how the proposed model differs from prior work, both in theoretical foundations and applied aspect.
>
> We agree with the reviewer that we did not make the difference with prior work clear and appreciate this being pointed out. The current paper is an extension of the previous work by [Ho et al., 2022, 2023], where the authors introduce a framework for modeling limited attention in humans (through value-guided construals). The previous work focused on simple discrete domains (gridworlds) to ensure tight experimental control for human experiments.  The current work extends that framework in three distinct ways: (1) we introduce heuristic construals, (2) we focus on developing a scalable algorithm for the inverse problem, and (3) we validate the approach in the substantially more complex domain of simulated driving in real-world scenarios. Our working manuscript now includes a more explicit discussion of how it differs from prior work in the introduction as well as background sections.
>
> > The paper would benefit from a dedicated section for experiments and results. Currently, the presentation of experiments is scattered, making it difficult to follow.
>
> > The manuscript lacks a Conclusion section. I suggest including one, even if brief, to summarize key findings and outline future work.
>
> We appreciate the reviewer’s comments on the clarity of the presentation of our work, for both experiments and summarization of the key findings. Our working manuscript has been reorganized. The experiments and results section now contains three experiments. Inference of heuristic parameters in gridworld, comparison with IRL, and Inference of heuristic parameters for GPUDrive. Additionally, a dedicated conclusion section has now been added to the manuscript that summarizes the main contributions of the paper. Our comments about potential future work have been added to a section on “Limitations and Future Work”
>
> > Figure 2 is difficult to interpret. There does not appear that the caption explaining the figure or how to extract information from it. Additionally, it is unclear how important it is for the reader to understand the estimated heuristic weights in tabular MDPs. Finally, the meaning of the values shown within the figure is not explained.
>
> We have updated the caption for the figure, it now says:
> Plots for model fit results for AAIP: actual weights of heuristic biases plotted against
> model estimates, in a simple Gridworld MDP. We simulated the behavior of 1000 agents (with different combinations of weights for three heuristics) by sampling 125 trajectories from 25 different DrivingWorld scenarios for each agent. Then, we attempted to learn the weights from behavior via maximum-likelihood estimation. Each sample represents the true weight of a heuristic for an agent and its estimated value derived from the agent’s behavior. Correspondence between true and estimated weights demonstrates that heuristics that are less consequential (e.g., a bias to ignore cones) are more readily identifiable from behavior (indicated by higher R-squared values). In contrast, biases that could be more consequential (e.g., a bias to ignore parked cars) are less identifiable from behavior because they are outweighed by the effect of value-guided construals. These results illustrate, in a simple setting, the viability and challenges of attention-aware inverse planning.
>
>
> > Figure 4 is not sufficiently descriptive. The arrows and annotations do not further clarify the context or content depicted, making it difficult to interpret the figure.
>
> Arrows have been removed, and markers have now been added to draw attention to the various aspects of the figures. In addition, the caption has been updated to:
> The three figures demonstrate the behavior (driving trajectory) of our PPO agent controlling the ego vehicle under three conditions: (i) when it can see every vehicle in the scene (left); (ii) when it can see a single vehicle that is relevant to the driving plan (middle); and (iii) when it can a single vehicle that is irrelevant to the driving plan (right). The ego trajectories in translucent red represent the behavior of the PPO agent across 40 trials in each condition. Solid lines of all other colors represent the trajectories of other vehicles in the scene. The purple solid lines in the middle and right plots show the trajectory of the single vehicles being observed in each condition. The behavior of our PPO agent (demonstrated by the distribution of ego trajectories) in the optimal condition (left) is similar to the condition where the agent’s construed state space contains a vehicle that is relevant to its plan (middle), as opposed to the condition where it observes a different irrelevant vehicle (right).

---

> > ### Comment · Reviewer_kdHu · 2025-08-05
> >
> > I appreciate the authors’ rebuttal and their efforts to address my concerns. I have updated my score accordingly.

---

### Official Review · Reviewer_qcxq · 2025-06-29

**Clarity:** 2
**Significance:** 3
**Originality:** 2
**Rating:** 4
**Confidence:** 2

**Summary:**

This paper introduces attention-aware inverse planning (AAIP), a framework for estimating people’s task-specific attentional bias from observed behavior by introducing an inverse problem of the value-guided construal model from previous work [1]. The authors (1) define the inverse problem of inferring bias parameters from state-action trajectories, (2) provide an MLE-based inference algorithm for this problem, (3) validate it in a tabular gridworld with accurate biases recovery performance, and show standard inverse reinforcement learning (IRL) fails to capture attention-limited policies, and (4) scale up to driving scenes simulated from the Waymo Open Dataset using a GPUDrive simulator and show accurate bias parameters recovery.

[1] Ho et al., People construct simplified mental representations to plan, Nature (2022)

**Questions:**

1. Given that AAIP currently assumes a known reward function, how might the framework be extended to infer both the reward and the attentional‐bias parameters λ when the reward is unknown or only partially specified?
2. Could you provide the time complexity and empirical runtime of AAIP?
3. Could you elaborate on why error bars and significance tests are unnecessary for experiments in this paper?

**Ethical Concerns:**

["NO or VERY MINOR ethics concerns only"]

**Final Justification:**

Since the authors have adequately addressed my concerns over extending AAIP to inferring reward function and empirical runtime, and (claimed to) have performed additional experiments regarding statistical significance & bias recovery performance and revised the manuscript accordingly, I increase my score to 4 (Borderline Accept).

**Limitations:**

yes

**Paper Formatting Concerns:**

No paper formatting concerns

**Quality:**

3

**Strengths And Weaknesses:**

**Strengths**
1. This paper clearly defines the attention-aware inverse planning problem and explains the motivation behind proposing this new problem (human behavior often has attention-related bias that cannot be captured by the standard IRL model). The problem proposed can be viewed as an inverse problem of bias parameters in the value-guided construal model from previous work [1].
2. This paper conducted experiments in both tabular and scalable setups, showcasing the accurate biases recovering performance of the proposed algorithm. In the tabular case, it also shows that standard IRL fails to capture attention-limited policies, emphasizing the need of the proposed AAIP framework to accurately model agent behavior.

**Weaknesses**
1. The AAIP framework treats the reward as a fixed, known input rather than inferring it from data. In contrast, standard IRL recovers both reward and policy. Relying on a predefined reward limits AAIP's applicability in complex settings where the true reward structure is unknown or difficult to specify.
2. The identifiability of bias in AAIP remains unclear. While the paper proposes a maximum‐likelihood estimator, it offers no analysis on when and to what degree the bias weights can be uniquely recovered from behavior. Empirically, both experiments explore only very simple bias spaces, so it is unknown how inference would perform in more complex settings. Even in the scalable Waymo experiment, the learnable bias parameter λ lies on a small discrete 1D space.
3. There is no discussion of the algorithm’s time complexity, nor empirical runtime report. Such discussion would help clarify computing resource requirements and efficiency.
4. No statistical significance info is provided for experiments. While authors argue that error bars and significance tests are unnecessary for this paper in the checklist, IRL literature typically reports results of multiple runs with different random seeds. Including such statistics would help readers understand the robustness of the reported bias‐recovery performance.
5. Although the inference method is described in the text, there is no pseudocode or algorithm block to represent the overall inference procedure. A clear, step‐by‐step outline, which specifies inputs, outputs, and optimization steps, would make the procedure easier to understand.

Overall, this paper introduces an interesting inverse problem of value-guided construal models that has potential in cognitive science research. However, I feel it is not yet ready for publication in its current form, given the limited scope of experiments and theoretical analysis.

---

> ### Author Rebuttal · Authors · 2025-07-31
>
> We thank the reviewer for their interest in our work and their thoughtful comments. We have thoroughly considered their feedback about the limitations of our work and have attempted to address them in the working version of the manuscript. Below we respond to the questions and elaborate on additional points raised.
>
> > Q1 Given that AAIP currently assumes a known reward function, how might the framework be extended to infer both the reward and the attentional‐bias parameters λ when the reward is unknown or only partially specified?
>
> > [related comment] The AAIP framework treats the reward as a fixed, known input rather than inferring it from data. In contrast, standard IRL recovers both reward and policy. Relying on a predefined reward limits AAIP's applicability in complex settings where the true reward structure is unknown or difficult to specify.
>
> We thank the reviewer for this question (and related comment). While we believe that AAIP is an interesting problem even when rewards are known, we entirely agree that inferring rewards is an important extension of our current approach. Indeed, it can be done; Rane et al. (2023) demonstrated that joint Bayesian inference of rewards and construals better capture human judgments than simply estimating rewards from behavior using IRL models. Note though that that work differs from ours in several ways—it was done with simple gridworlds, focused on comparison with human judgments, and did not use *value-guided* construals.
>
> In the context of the current work, we can sketch out how joint inference over heuristic parameters and rewards should be specified. Given observed behavior $\zeta = \langle s_1, a_1, …, s_T, a_T \rangle$, the posterior over reward function parameters $\theta$ in Bayesian IRL is:
>
> $$
> P(\theta \mid \zeta) \propto P(\theta) \prod_{t} \pi^*_\theta(a_t \mid s_t)
> $$
>
> where $\pi^*_\theta$ is the optimal policy in the task under a reward parameterization $\theta$. Jointly inferring rewards and attention parameters $\lambda$ would have the following posterior:
>
> $$
> P(\theta, \lambda \mid \zeta) \propto P(\theta)P(\lambda) \sum_c \pi_{\lambda}(c \mid s) \prod_t \pi^*_{\theta}(a_t \mid s_t[c])
> $$
>
> Where $\pi_{\lambda}(c \mid s)$ is from Equation 3 in the original submission. Note that this is a more complex inference problem since the construal is a hidden state that persists over the course of the trajectory. Indeed, we speculate that this joint inference can only be robustly solved if treated in a fully Bayesian manner—that is, integrating over possible $\theta$ and $\lambda$ and not simply solving for MLE parameters. This is an important direction that we hope will be explored more thoroughly in future work, and in the discussion now write “The current work focuses on AAIP given a known reward, but in many domains rewards are unknown or individuals may have idiosyncratic preferences. Thus, the applicability of AAIP will be greatly expanded by developing new methods that allow for joint inference of attentional heuristics and rewards.”
>
> > Q2 Could you provide the time complexity and empirical runtime of AAIP?
>
> > [related comment] There is no discussion of the algorithm’s time complexity, nor empirical runtime report. Such discussion would help clarify computing resource requirements and efficiency.
>
> We thank the reviewer for raising this point.
>
> Prior to inference, our algorithm requires evaluating construals to calculate their behavioral utility and therefore their probability of being chosen. This means there are two main computational bottlenecks: (1) computing an optimal policy and (2) evaluating an optimal policy in the true task using a simulator. One of the algorithmic contributions of our paper is a more efficient way to compute the optimal policy, which essentially involves amortizing the cost by pre-training a generalist policy using standard RL methods (e.g., PPO on the GPUDrive platform). This can be done on a standard GPU within a day (note GPUDrive includes a pre-trained generalist policy that we used). The *evaluation stage* requires simulating a masked policy on the true dynamics. This took an hour to evaluate 166 construals across 10 Waymo scenarios by sampling 40 trajectories per construal. We were only able to partially parallelize the evaluation process due to limitations in current implementation of the Python interface, but we note that evaluation can be significantly sped up by optimally parallelizing the process in GPUDrive. At inference time, it takes about 3 min and 30 sec to perform inference over 80 trajectories (roughly 12 minutes of driving data) for a single set of parameters (machine details: Intel Core Ultra 7 165H laptop CPU with 32GB of RAM). We will release all code used in the project once the paper is published and include information about approximate runtime.
>
> In addition, we now include results related to the real-time data efficiency of an updated version of our algorithm. The updated algorithm can infer multiple continuous heuristic parameters jointly using gaussian process optimization (the original manuscript reported an experiment with only a single parameter naively optimized over discrete values). In the working manuscript, we now report results for estimating 3 continuous heuristic parameters. To evaluate real-time data efficiency on the Waymo scenarios in the GPUDrive simulator, we evaluated the algorithm with different amounts of driving data, ranging from the equivalent of 1.5 minutes of driving to 75 minutes of driving. With 1.5 minutes of driving data, the mean squared error between true and estimated parameters is 52.64 (averaged over 30 simulations). At 12 minutes of driving data this decreases to 19.56 and stays around this value with more data. A plot of these results is now included in the supplement of the manuscript (we did not include it here due to the rebuttal rules).
>
> > Q3 Could you elaborate on why error bars and significance tests are unnecessary for experiments in this paper?
>
> > [related comment] No statistical significance info is provided for experiments. While authors argue that error bars and significance tests are unnecessary for this paper in the checklist, IRL literature typically reports results of multiple runs with different random seeds. Including such statistics would help readers understand the robustness of the reported bias‐recovery performance.
>
> We did not include error bars or significance tests since we plotted the raw data and the results appeared visually robust, but in our working manuscript we now include error bars and significance tests for old and new analyses. Specifically, since our submission, we have conducted more extensive GPUdrive experiments in which we attempt to jointly estimate 3 heuristics in the scenarios, similar to the DrivingWorld paradigm. For both sets of experiments, we now report significance tests to compute the R-squared value for the model fit (actual vs. estimated heuristic parameters) and p-value measures for both model fits. The P-values for the GPUDrive experiment are in the neighborhood of 1.0E-80. For our gridworld experiment, p-values range around 1.0E-300.
>
> > The identifiability of bias in AAIP remains unclear. While the paper proposes a maximum‐likelihood estimator, it offers no analysis on when and to what degree the bias weights can be uniquely recovered from behavior. Empirically, both experiments explore only very simple bias spaces, so it is unknown how inference would perform in more complex settings. Even in the scalable Waymo experiment, the learnable bias parameter λ lies on a small discrete 1D space.
>
> We acknowledge more work could be done to analyze the degree to which bias weights can be recovered. In new work, now included in the updated manuscript, we’ve extended the Waymo experiment, where we perform joint inference over a continuous 3D heuristic space. Our results show excellent recoverability with R-squared values 0.77, 0.76, and 0.83 (p < $10^{-4}$) for each of the three complex heuristics even with a modest amount of data (the equivalent of roughly 12 minutes of real-time driving). Further, we discuss specific avenues for moving to larger bias spaces or learning them in our updated discussion.
>
> > Although the inference method is described in the text, there is no pseudocode or algorithm block to represent the overall inference procedure. A clear, step‐by‐step outline, which specifies inputs, outputs, and optimization steps, would make the procedure easier to understand.
>
> We now include an algorithm block in our paper to lay out our complete inference process.

---

> > ### Comment · Reviewer_qcxq · 2025-08-04
> >
> > Thank you for your thoughtful response! Since the authors have adequately addressed my concerns over extending AAIP to inferring reward function and empirical runtime, and (claimed to) have performed additional experiments regarding statistical significance & bias recovery performance and revised the manuscript accordingly, I will increase my score to 4 (Borderline Accept).

---

### Official Review · Reviewer_MaQm · 2025-07-02

**Clarity:** 4
**Significance:** 3
**Originality:** 3
**Rating:** 4
**Confidence:** 4

**Summary:**

This manuscript introduces a framework called Attention-Aware Inverse Planning (AAIP) to interpret human cognitive limitations based on their behaviors. The core assumption is that people can’t plan with all the sensory inputs they receive from the external world due to perceptual costs, and thus they plan with a limited subset of objects in that state space. Unlike Inverse Reinforcement Learning (IRL), the framework does not assume a special form of reward function for these sub-state decision makers. The model aims to work backward from an observed behavior, such as a driving trajectory, to estimate the specific parameters of cognitive biases that have caused such limitations. The authors test this approach using simulated agents in a simple grid world and a more realistic driving simulator, showing that their method can successfully recover the built-in biases, unlike IRL.

**Questions:**

1. In lines 214-215, the author used a “known” reward function for IRL, and with this reward design, the AAIP outperforms IRL in later experiments. However, I wonder if we can sample a reward function from more expressive reward function spaces; will things be different? Or, to put it more generally, I wonder if the deficit of IRL, which assumes full observation of states, can always be compensated by reward function design. Is there any bridge that can unify these two frameworks? Or is there an impassable divide between them? For example, can we borrow ideas from Partially Observable MDP to address the problem of vanilla IRL? These questions go far beyond the scope of the current paper, but since one of the core experiments in this study involves comparing two frameworks, I think a thorough discussion with some quantitative analysis would make the results far more convincing.

2. Similar issues like Q1 regarding the specific form of cognitive costs and heuristic feature engineering exist in the manuscript. I expect more discussion on this in the final sections.

**Ethical Concerns:**

["NO or VERY MINOR ethics concerns only"]

**Final Justification:**

I have read the rebuttal and found it partially addresses my concerns. I recommend accepting this paper.

**Limitations:**

1. The study relies entirely on synthetic data and lacks validation with actual human behavior. The authors allude to this as a future step but don't frame it as a direct limitation of their current work.
2. The framework cannot discover new cognitive biases; they must be defined by researchers beforehand. This makes it implausible in end-to-end cases where biases must be inferred automatically. Instead, the authors claim that more “complex heuristics” will be used in those cases, which I don’t see as a good choice for real-world problems.
3. The model's focus is narrowly restricted to sub-optimality from inattention, ignoring other key factors in human decision-making, such as risk aversion/preference or memory constraints. The authors do not discuss this limitation.

**Paper Formatting Concerns:**

no such concerns from my side

**Quality:**

2

**Strengths And Weaknesses:**

**Strengths**

1.  AAIP is  a new and important way to model behavior by inferring cognitive biases, moving beyond the limitations of traditional IRL, which might be crucial for understanding human behaviors and further design machines that interact with humans.
2. The model is well-grounded in established cognitive theories like bounded rationality and value-guided construal, giving it strong theoretical validity and explanatory power.

**Weaknesses**

1. The paper's primary weakness is its exclusive reliance on synthetic data. Without real human data that includes humans' cognitive biases through self-report or eye movement, it’s hard to verify the explanatory power of the framework.
2. The equation used to represent the value function assumes an addable state representation, which is not true in many real-world settings because the actual value in a task may rely on more than two-order level relationships of objects. This is also a core issue in modern DL interpretability studies where the inputs are high-dimensional, like images and videos. Therefore, I doubt if this framework can be applied to problems that are solved in an end-to-end fashion.
3. The framework makes very simplified assumptions about decision makers’ attention, that they are simply selecting a fixed group of objects in the whole episode. This is a significant simplification that doesn't reflect the dynamic nature of attention in most real-world situations.

---

> ### Author Rebuttal · Authors · 2025-07-31
>
> We thank the reviewer for their interest in our work and their helpful comments. Below we first respond to the questions and then elaborate on additional points raised.
>
> > Q1 In lines 214-215, the author used a “known” reward function for IRL, and with this reward design, the AAIP outperforms IRL in later experiments. However, I wonder if we can sample a reward function from more expressive reward function spaces; will things be different? Or, to put it more generally, I wonder if the deficit of IRL, which assumes full observation of states, can always be compensated by reward function design. Is there any bridge that can unify these two frameworks? Or is there an impassable divide between them? For example, can we borrow ideas from Partially Observable MDP to address the problem of vanilla IRL? These questions go far beyond the scope of the current paper, but since one of the core experiments in this study involves comparing two frameworks, I think a thorough discussion with some quantitative analysis would make the results far more convincing.
>
> We thank the reviewer for raising these important questions about the relationship between IRL and AAIP that get to the heart of our work. The simulations in section 3.5 were designed as a simple quantitative case study to show some of the limits of IRL, but we completely agree that an exciting direction for future work is a more formal analysis of these two paradigms.
>
> In the discussion, we now highlight work that we believe sheds light on the relationship between IRL and AAIP. Specifically, theoretical results in “Robust IRL” show that if the transition function of an IRL learner is misspecified, this affects the bound on the gap between the value of the learned policy and the true policy (https://arxiv.org/pdf/2007.01174). Applying IRL to an attention-limited decision-maker’s actions is similar in that such a decision-maker is acting wrt a simplified transition model. Thus, we know in principle that there can be a recoverability gap when doing IRL in general. It is an open question how pervasive this gap is in real-world driving scenarios or other tasks, but the simple example in section 3.5 suggests it is not difficult to construct such scenarios.
>
> > Q2 Similar issues like Q1 regarding the specific form of cognitive costs and heuristic feature engineering exist in the manuscript. I expect more discussion on this in the final sections.
>
> We agree that it is important to extend this work to learning cognitive costs and heuristics from data. In the discussion of the manuscript, we now include the following sentence: “Similar to how we can infer the heuristic $\lambda$ weights from data, we could also infer parameters of the heuristic feature functions themselves $\{\phi\}$ from data from the gradient of the likelihood, though this would be more data-intensive. Future work should explore this possibility to extend the generality of our proposed framework.”
>
> > The equation used to represent the value function assumes an addable state representation, which is not true in many real-world settings because the actual value in a task may rely on more than two-order level relationships of objects.
>
> This is an important comment that echoes a point made by R1. **We did not make clear in the original submission that the value functions computed by the model does take into account higher-order object relationships**. We include our explanation here for the reviewer’s convenience:
>
> Although the construal state $s[c]$ is modeled as a set of objects, the plan computed with respect to $s[c]$ **takes the inter-object relationships and spatio-temporal dynamics into account through the transition function**. For example, suppose we have a state with objects {$o_1, o_2, o_3, o_4, o_5$}, but our construal is $c =$  {$o_1, o_2, o_3$}. The optimal policy $\pi_{c}(a \mid s)$ is in fact being computed wrt dynamics that incorporate all the interactions between $o_1$, $o_2$, and $o_3$ while ignoring anything having to do with $o_4$ or $o_5$.
>
> The confusion comes from the fact that in section 3.2, we jumped to the generalist policy formulation without elaborating on the more basic formulation. In the original value-guided construal formulation, an optimal policy for a construal $c$ is associated with a value function with respect to a construed reward function and task, i.e.:
>
> $Q_{c}(s, a) = R_{c}(s, a) + \gamma\sum_{s’} T_{c}(s’ \mid s, a) V_c(s’)$
>
> The reward function $R_c$ and transition function $T_{c}$ incorporate the interactions between objects in $c$ (and nothing else). This means the value functions and policy (i.e., $V_c$, $Q_c$, and $\pi_c$) also incorporate information about interactions. In the generalist policy formulation, rather than express separate $R_c$ and $T_c$ for each construal $c$, we assume we have $R$ and $T$ that can operate over masked states, $s[c]$, such that $R_c(s, a) = R(s[c], a)$ and $T_c(s’ \mid s, a) = T(s’[c] \mid s[c], a)$. This assumption on $R$ and $T$ makes things notationally convenient (fewer subscripts) and also allows the use of a pre-trained generalist policy described later in the paper. We remarked upon this assumption in section 3.1 but we did not make the connection explicit.
>
> In our working manuscript we have added these details to clarify that the construal state does indeed incorporate inter-object interactions.
>
> > The framework makes very simplified assumptions [...] that doesn't reflect the dynamic nature of attention in most real-world situations.
>
> We agree and acknowledge that this is an important next step, and consequently, address this limitation in the discussion as a problem for future work. However, we also want to point out that the degree to which this assumption is problematic depends on the timescale of trajectories. Individual trajectories in the Waymo dataset represent 9 seconds of real-time behavior, which could limit how much dynamic attention is occurring. Additionally, in previous psychology work on value-guided construals done with maze navigation, it was found that a single fixed construal approximated the “cumulative construal” when attentional processes were modeled dynamically (Ho et al., 2022; supplemental material). In other words, the set of things one paid attention to over the course of an entire trial was similar to the set of things one would pay attention to all at once, if they had to. While this still results in a loss of information (i.e., precisely when someone paid attention to something), it shows that assuming a single fixed construal is a reasonable simplification for short timescales (~5-10 seconds). Acknowledging the caveat that these were experiments in mazes and not in a driving domain, this suggests that assuming that drivers select a fixed group of objects across a 9 second episode is a reasonable simplification.
>
> > The study relies entirely on synthetic data and lacks validation with actual human behavior. The authors allude to this as a future step but don't frame it as a direct limitation of their current work.
>
> We thank the reviewer for pointing out this oversight and now explicitly state this limitation of the current work. In the discussion we have now added “Finally, a key limitation of the current work is that it focuses on validation on simulated data and not human behavior.”
>
> > The framework cannot discover new cognitive biases; they must be defined by researchers beforehand. This makes it implausible in end-to-end cases where biases must be inferred automatically. Instead, the authors claim that more “complex heuristics” will be used in those cases, which I don’t see as a good choice for real-world problems.
>
> We agree with the reviewer’s comment that the current paper does not discover new biases. Part of the reason for this is that we would like our framework to allow users to evaluate pre-specified or hypothesized biases, which is useful in cognitive science applications as well as other domains where interpretability is important. That said, we believe that our framework can straightforwardly accommodate inferring new attentional biases by allowing the $\phi$ functions (described in section 3.3) to be parameterizable and learned from data with gradient-based methods. The main challenge with doing this is that it would require larger data sets, but we see our approach as compatible with data driven methods and see this as an exciting direction for future work.
>
> Additionally, in relation to the comment about end-to-end learning, we would like to clarify that while our approach does require pre-processing (e.g., extracting object representations), we take this approach intentionally since it allows us to work in settings where data is less abundant or the internal representations used by an algorithm are important. This includes work on autonomous driving as well as human-machine interaction (e.g., semi-automated driving, human-machine teaming, machine teaching).
>
> > The model's focus is narrowly restricted to sub-optimality from inattention, ignoring other key factors in human decision-making, such as risk aversion/preference or memory constraints. The authors do not discuss this limitation.
>
> We agree with the reviewer that the current paper focuses on sub-optimality from attention and does not incorporate other kinds of biases. Our framework can be extended to incorporate these additional factors, as we discuss in the new limitation section (pasted below):
>
> Although we focus on one cognitive dimension of suboptimal driving behavior, other psychological factors, such as propensity for risky taking, ability of risk perception, and slow response times, can also lead to suboptimal decisions in driving tasks. It is possible to incorporate other psychological factors into our model in the future. For instance, joint inference of heuristic biases and reward functions would allow for simultaneously estimating a driver's underlying heuristics and tendency for risky behavior.

---

> > ### Comment · Reviewer_MaQm · 2025-08-05
> >
> > I thank the authors for their thorough response to my questions. Regarding Q1, which asks about the limits of IRL with better or more diverse reward function space design, the authors claimed that [this paper](https://arxiv.org/pdf/2007.01174) is one exploration that shows the gap. However, I was originally pointing to more theoretical bridges, such as the existence of such a possibility rather than empirical studies, with which one can only show things case by case, and the limitation shown cannot deny the existence. But this is quite off-topic. I thank them for providing extra information.
> >
> > For addable state representations, the added description now makes it clear. However, I still wonder how this exploding approach could scale up to real-world scenarios where even objects are hard to define from modalities like vision. Maybe something like a discrete information bottleneck could do the job.
> >
> > The further discussion about human attention scale makes sense to me.
> >
> > Based on the discussion, I decide to keep my score the same as the original.

---

### Official Review · Reviewer_3HcF · 2025-07-03

**Clarity:** 3
**Significance:** 3
**Originality:** 3
**Rating:** 5
**Confidence:** 2

**Summary:**

This paper introduces attention-aware inverse planning, aiming to infer a person's attentional biases from their behavior. Unlike standard IRL, which assumes near-optimal decision-making, this work models how limited attention shapes choices using a parameterized bias function grounded in the value-guided construal framework. The authors also propose a scalable algorithm combined with deep RL and cognitive modeling to learn these biases via maximum likelihood. Experiments on show that this method outperforms IRL in capturing attention-limited behaviors in simulation experiments.

**Questions:**

What is the latency of the proposed method?

**Ethical Concerns:**

["NO or VERY MINOR ethics concerns only"]

**Final Justification:**

The authors addressed my concerns, I recommend accept.

**Limitations:**

See weakness.

**Quality:**

3

**Strengths And Weaknesses:**

I like the motivation of this work. Specifically, the authors address the gap between cognitive science findings and standard ML models. This is interesting to me especially about the realistic modeling framework grounded in human cognition. Besides, the proposed method provides a systematic, extensible approach for explaining, predicting, and influencing deviations from ideal rational behavior. Experiments also demonstrate the effectiveness of the proposed methods with outperforming standard IRL.

However, the attentional heuristics used are relatively simple. The authors also mentioned this. Capturing more nuanced real-world biases may require richer heuristic models. Is there some better ways? Besides, although the method is claimed to be scalable and is demonstrated on the Waymo Open Dataset, the inference process—particularly the marginalization over all possible construals (as in Eq. 6)—can be computationally expensive in theory, raising concerns about scalability to more complex domains. It would be better for the authors to provide some latency analysis.

---

> ### Author Rebuttal · Authors · 2025-07-31
>
> We thank the reviewer for their interest in our work!
>
> > Q1 What is the latency of the proposed method?
>
> Our current implementation performs inference in Python on a CPU since this gives us the fine-grained control of the various components. Further, inference over various construals are currently performed sequentially; processing time can be reduced significantly through parallelization. Even using this non-optimized version, we are able to perform inference (estimate heuristic parameters for a trajectory of actions) for the equivalent of 12 minutes of driving data in 3 minutes and 30 seconds.
>
> Additionally, since our original submission, we have performed more systematic analysis of the performance of our algorithm in terms of its real-time data complexity. The updated algorithm can infer multiple continuous heuristic parameters jointly using gaussian process optimization (the original manuscript reported an experiment with only a single parameter naively optimized over discrete values). In the working manuscript, we now report results for estimating 3 continuous heuristic parameters. To evaluate real-time data efficiency on the Waymo scenarios in the GPUDrive simulator, we evaluated the algorithm with different amounts of driving data, ranging from the equivalent of 1.5 minutes of driving to 75 minutes of driving. With 1.5 minutes of driving data, the mean squared error between true and estimated parameters is 52.64 (averaged over 30 simulations). At 12 minutes of driving data this decreases to 19.56 and stays around this value with more data. A plot of these results is now included in the supplement of the manuscript (we did not include it here due to the rebuttal rules).

---

> > ### Comment · Reviewer_3HcF · 2025-08-09
> > **Addressed my concerns**
> >
> > Thanks for the rebuttal. The authors addressed my concerns.

---

### Official Review · Reviewer_tfig · 2025-07-06

**Clarity:** 3
**Significance:** 3
**Originality:** 4
**Rating:** 5
**Confidence:** 3

**Summary:**

This paper highlights the impact of human cognitive biases and introduces attention-aware inverse planning, a method for inferring attentional biases from human behavior. It distinguishes this approach from standard inverse reinforcement learning and presents an algorithm that integrates deep reinforcement learning with cognitive modeling. Applied to driving data from the Waymo Open Dataset, the method uncovers drivers’ attentional strategies.

This paper makes a meaningful contribution to understanding and modeling human cognitive biases in decision-making. Despite some limitations in representation and modeling complexity, the framework is well-motivated, novel, and reasonably validated. I recommend acceptance.

**Questions:**

+ Could attentional biases be directly incorporated into the inverse reinforcement learning framework? For example, [Predicting Goal-directed Human Attention Using Inverse Reinforcement Learning, Zhang et al - CVPR 2020] model goal-directed attention using IRL.

+ Is the construal mechanism conceptually related to personalized saliency or scanpath prediction? For instance, Xue et al. (Few-shot Personalized Scanpath Prediction - CVPR 2025) propose a method for few-shot personalized scanpath prediction, claiming the ability to learn individualized scanpaths with only a few episodes.

**Ethical Concerns:**

["NO or VERY MINOR ethics concerns only"]

**Final Justification:**

I have read the rebuttal and found it addresses my concerns quite well. I recommend accepting this paper.

**Limitations:**

Yes

**Paper Formatting Concerns:**

None noted.

**Quality:**

4

**Strengths And Weaknesses:**

Strengths:

+ Clarity and Motivation: The paper is well-written and clearly motivated, with a strong theoretical foundation and relevant citations.

+ Novelty and Relevance: The focus on cognitive biases and limited attention in human decision-making is timely and important.

+ Framework Design: The proposed framework is conceptually simple yet general, combining reinforcement learning components (reward, value, policy) with heuristics-based modeling of attentional biases.

+ Scalability: Application to large-scale driving data showcases the method’s practical viability and relevance to real-world problems.

Weaknesses:

+ Simplified Construal Representation: The construal state is modeled as a set of objects, without accounting for inter-object relationships or spatial-temporal dynamics, which may limit expressiveness.

+ Pipeline Complexity: The framework is not end-to-end; it relies on object detection as a preprocessing step to construct the construal space. This may introduce dependencies and limit generalizability.

+ Heuristic Modeling: the heuristics used to uncover the cognitive biases are relatively simple and may not capture richer cognitive phenomena such as relational attention or dynamic prioritization.

---

> ### Author Rebuttal · Authors · 2025-07-31
>
> We are grateful to the reviewer for their interest in our work and their thoughtful comments. Below we address the main questions that were brought up and respond to some of the weaknesses noted:
>
>
> > Q1 Could attentional biases be directly incorporated into the inverse reinforcement learning framework? For example, [Predicting Goal-directed Human Attention Using Inverse Reinforcement Learning, Zhang et al - CVPR 2020] model goal-directed attention using IRL.
>
> We thank the reviewer for pointing us to the paper by Zhang et al (2020), which models scanpaths (i.e., gaze patterns) as goal-directed by learning an eye fixation policy with IRL. The types of tasks that they applied their method to are ones in which people are shown an image and must find a target object (e.g., identify a fork in a scene of a dinner table). Identifying certain objects in a scene is an important subtask in more complex tasks such as driving (e.g., to decide if you can cross an intersection, you need to identify the streetlight, among other things). Thus, the attentional biases learned by Zhang et al’s IRL method could be used to extract heuristics that could be used in our AAIP method (or a method that combines IRL with AAIP). This is an interesting direction that builds on the previous results in the scanpath literature that we were previously unaware of. In the working manuscript, we now reference Zhang et al.’s paper as well as discuss how such methods could be used in concert with our approach.
>
> > Q2: Is the construal mechanism conceptually related to personalized saliency or scanpath prediction? For instance, Xue et al. (Few-shot Personalized Scanpath Prediction - CVPR 2025) propose a method for few-shot personalized scanpath prediction, claiming the ability to learn individualized scanpaths with only a few episodes.
>
> We appreciate the reviewer raising this connection to Xue et al. (2025). Xue et al develop a novel algorithm for learning individualized scanpaths (gaze patterns) efficiently from subjects who are freeviewing or searching images. The construal mechanism overlaps with this approach to gaze modeling in that both are models of attention. However, a key difference is that the construal mechanism provides an account of how attention is affected by the need to plan—it models what someone would pay attention to if they needed to plan a sequence of actions in a certain situation. Xue et al. show how you can efficiently learn personalized scanpaths in a setting that does not involve planning, but their general approach could provide a template for also modeling construals. We see two main challenges to applying Xue et al.’s particular method to the problem of value-guided construal: (1) incorporating a notion of planning into their architecture, and (2) modeling attention and/or gaze as an unobserved variable that is marginalized over since in our setting we do not assume access to gaze data. As with the Zhang et al. paper, this is an interesting connection to the existing literature that we had not considered and now reference in both the Background and future directions.
>
> > Simplified Construal Representation: The construal state is modeled as a set of objects, without accounting for inter-object relationships or spatial-temporal dynamics, which may limit expressiveness.
>
> This comment as well as R3’s point about addable state representations highlights an important feature of our model that we did not make clear in the original submission. We apologize for this confusion and hope to clarify it here.
>
> Although the construal state $s[c]$ is modeled as a set of objects, the plan computed with respect to $s[c]$ **takes the inter-object relationships and spatio-temporal dynamics into account through the transition function**. For example, suppose we have a state with objects {$o_1, o_2, o_3, o_4, o_5$}, but our construal is $c =$  {$o_1, o_2, o_3$}. The optimal policy $\pi_{c}(a \mid s)$ is in fact being computed wrt dynamics that incorporate all the interactions between $o_1$, $o_2$, and $o_3$ while ignoring anything having to do with $o_4$ or $o_5$.
>
> The confusion comes from the fact that in section 3.2, we jumped to the generalist policy formulation without elaborating on the more basic formulation. In the original value-guided construal formulation, an optimal policy for a construal $c$ is associated with a value function with respect to a construed reward function and task, i.e.:
>
> $Q_{c}(s, a) = R_{c}(s, a) + \gamma\sum_{s’} T_{c}(s’ \mid s, a) V_c(s’)$
>
> The reward function $R_c$ and transition function $T_{c}$ incorporate the interactions between objects in $c$ (and nothing else). This means the value functions and policy (i.e., $V_c$, $Q_c$, and $\pi_c$) also incorporate information about interactions. In the generalist policy formulation, rather than express separate $R_c$ and $T_c$ for each construal $c$, we assume we have $R$ and $T$ that can operate over masked states, $s[c]$, such that $R_c(s, a) = R(s[c], a)$ and $T_c(s’ \mid s, a) = T(s’[c] \mid s[c], a)$. This assumption on $R$ and $T$ makes things notationally convenient (fewer subscripts) and also allows the use of a pre-trained generalist policy described later in the paper. We remarked upon this assumption in section 3.1 but we did not make the connection explicit.
>
> In our working manuscript we have added these details to clarify that the construal state does indeed incorporate inter-object interactions.
>
> > Pipeline Complexity: The framework is not end-to-end; it relies on object detection as a preprocessing step to construct the construal space. This may introduce dependencies and limit generalizability.
>
> The reviewer is correct that the current method is not end-to-end and currently requires object detection as a preprocessing step. However, we would like to note that this approach is widely used in work on autonomous driving as well as human-machine interaction (e.g., semi-automated driving, human-machine teaming, machine teaching). For example, perception stacks in autonomous driving typically include object detection (https://arxiv.org/abs/2106.12735) and tracking (https://arxiv.org/abs/2101.02702). Additionally, this approach is useful since it allows us to work in settings with sparser data and provides more control over the internal representations the algorithm uses.
>
> In the case of human-machine interaction, we especially care about having algorithms that produce interpretable representations and also for those algorithms to accurately reason about people’s internal representations. People reason about the world in terms of objects and their interactions (“Functional Significance of Visuospatial Representations”, Tversky, 2005), and to learn similar kinds of representations, an end-to-end system would need much more data than is typically available in human interaction settings. Moreover, even if such data were available, there is unfortunately no guarantee that an end-to-end system would form similar representations to people, which is often important for interactive systems that reason about human decisions. That said, we view our approach as compatible with more data-driven approaches and are currently exploring extensions in which components like the attentional heuristics that are currently presented to the model can be learned from data.
>
> > Heuristic Modeling: the heuristics used to uncover the cognitive biases are relatively simple and may not capture richer cognitive phenomena such as relational attention or dynamic prioritization.
>
> We agree with the reviewer that more complex heuristics would be an exciting direction for future work and include a discussion of this in the working manuscript. Here, we address some of the reviewer’s concerns about psychological phenomena explicitly mentioned: (1) dynamic prioritization could be implemented as a bottom-up process where changes in object saliency, or significant changes in ego state could trigger an attention re-allocation event. (2) relational attention may be incorporated by deconstructing larger multi-object construals, where values are computed individually for sub-construals that reflect different heuristic preferences, limiting relational dynamics of objects within each sub-construal.
>
> Additionally in our working manuscript, we explore two new complex heuristics (relative heading and collision trajectory), in follow up experiments with GPUDrive. We have results for joint inference of the three continuous space heuristics, where we show excellent recoverability of the heuristic parameters (R-squared: 0.77, 0.76, 0.83).

---

### Note · Authors · 2025-08-13

We thank the reviewers for their interest in our work and their valuable feedback, which has enabled us to substantially improve our paper while sharpening our contributions and highlighting exciting directions for future research. We are also grateful that the reviewers have increased their scores following our rebuttal. For the Area Chair’s convenience, we summarize the changes and main points of discussion below:

1. We now report results from followup experiments where we show that our approach can be used to perform joint inference over complex heuristics in a 3-D continuous space, demonstrating AAIP’s versatility in scaling up to complex higher-dimensional spaces. This addresses concerns brought up by reviewers tfig, and qcxq regarding the use of simple heuristics and inference procedures in our experiments.
2. We also report statistical significance values for all our experiments, demonstrating the reliability of our results, as pointed out by reviewer qcxq.
3. We now provide additional information about the data efficiency and the program execution time of our implementation of AAIP, in response to queries about program latency by reviewers 3HcF, and qcxq.
4. We have restructured the manuscript to ensure a clear separation of theory and experiments, as suggested by reviewer kdHu.
5. We have updated figures and captions in the manuscript to address reviewer kdHu’s concerns about their clarity.
6. We now include the pseudocode for our driving simulator inference algorithm in the appendix, based on reviewer qcxq’s suggestion.
7. Finally, we now clearly explain how our algorithm for computing construal values accounts for behavioral interaction between objects within a construal, through the use of construed reward and transition functions. This was done in response to concerns about our formulation of the construal value function, raised by reviewers tfig, and MaQm.

---

### Decision · Program_Chairs · 2025-09-17

**Decision:**

Accept (spotlight)

**Comment:**

The submission proposed an alternative to inverse reinforcement learning (IRL): find the rewards that explain a behaviour. In principle, IRL could be used to capture some forms of bias by agents. The submission proposes to do IRL, but instead of rewards, the bias is towards a subset of objects.

The reviewers agree on the elegance of the idea and the overall presentation.

Among the weaknesses, reviewers thought that focusing on objects might fail to capture relationships, but the authors clarified that point. Another point was that the limited experiments. The authors performed additional experiments.

The authors acknowledge that their method is not end-to-end as it requires object detection, and argue that this is very common in concrete applications and improves interpretability.

The consensus is an explicit acceptance.

Personally, I recommend that the authors clarify the technical similarities between the IRL formulation and the value-guided construal. Reducing aparent differences with IRL might increase the impact of your work.